# Sugar Starvation Disrupts Lipid Breakdown by Inducing Autophagy in Embryonic Axes of Lupin (*Lupinus* spp.) Germinating Seeds

**DOI:** 10.3390/ijms241411773

**Published:** 2023-07-21

**Authors:** Sławomir Borek, Szymon Stefaniak, Katarzyna Nuc, Łukasz Wojtyla, Ewelina Ratajczak, Ewa Sitkiewicz, Agata Malinowska, Bianka Świderska, Karolina Wleklik, Małgorzata Pietrowska-Borek

**Affiliations:** 1Department of Plant Physiology, Faculty of Biology, Adam Mickiewicz University Poznań, Uniwersytetu Poznańskiego 6, 61-614 Poznań, Poland; szymon.stefaniak@amu.edu.pl (S.S.); lukasz.wojtyla@amu.edu.pl (Ł.W.); karolina.wleklik@amu.edu.pl (K.W.); 2Department of Biochemistry and Biotechnology, Faculty of Agronomy, Horticulture and Bioengineering, Poznań University of Life Sciences, Dojazd 11, 60-632 Poznań, Poland; katarzyna.nuc@up.poznan.pl (K.N.); malgorzata.pietrowska-borek@up.poznan.pl (M.P.-B.); 3Institute of Dendrology, Polish Academy of Sciences, Parkowa 5, 62-035 Kórnik, Poland; eratajcz@man.poznan.pl; 4Mass Spectrometry Laboratory, Institute of Biochemistry and Biophysics, Polish Academy of Sciences, Pawińskiego 5a, 02-106 Warsaw, Poland; ewa@ibb.waw.pl (E.S.); esme@ibb.waw.pl (A.M.); bianka@mslab-ibb.pl (B.Ś.)

**Keywords:** asparagine, embryo, iTRAQ, lipid droplet, lipophagy, peroxisome, pexophagy, proteomics, transcriptomics, ultrastructure

## Abstract

Under nutrient deficiency or starvation conditions, the mobilization of storage compounds during seed germination is enhanced to primarily supply respiratory substrates and hence increase the potential of cell survival. Nevertheless, we found that, under sugar starvation conditions in isolated embryonic axes of white lupin (*Lupinus albus* L.) and Andean lupin (*Lupinus mutabilis* Sweet) cultured in vitro for 96 h, the disruption of lipid breakdown occurs, as was reflected in the higher lipid content in the sugar-starved (-S) than in the sucrose-fed (+S) axes. We postulate that pexophagy (autophagic degradation of the peroxisome—a key organelle in lipid catabolism) is one of the reasons for the disruption in lipid breakdown under starvation conditions. Evidence of pexophagy can be: (i) the higher transcript level of genes encoding proteins of pexophagy machinery, and (ii) the lower content of the peroxisome marker Pex14p and its increase caused by an autophagy inhibitor (concanamycin A) in -S axes in comparison to the +S axes. Additionally, based on ultrastructure observation, we documented that, under sugar starvation conditions lipophagy (autophagic degradation of whole lipid droplets) may also occur but this type of selective autophagy seems to be restricted under starvation conditions. Our results also show that autophagy occurs at the very early stages of plant growth and development, including the cells of embryonic seed organs, and allows cell survival under starvation conditions.

## 1. Introduction

During seed germination, storage compounds are mobilized, allowing seedling establishment and growth. The main storage compound of lupin seeds is protein, with the majority represented by globulins [1]. Some lupin species may accumulate storage protein at the level of up to 50% of seed dry matter. Examples of species producing high protein-storing seeds are yellow lupin and Andean lupin. Apart from proteins, lupin seeds also accumulate lipids. The lipid content of lupin species varies significantly from about 5–6% in seeds of yellow lupin, through 11–14% in seeds of white and narrow-leafed lupins, up to about 20% in seeds of Andean lupin. Mature and dry lupin seeds contain only trace amounts of starch and are considered non-starch seeds, but they contain quite a high amount of other carbohydrates, mainly oligosaccharides, reaching in total up to about 36%, including 26% fiber [2]. Although mature and dry lupin seeds do not contain starch, this polysaccharide appears in lupin seed organs, especially in embryonic axes, during seed imbibition and germination [3]. Due to the main storage compound of lupin seeds being protein, the metabolism of germinating lupin seeds is based on amino acid interconversions. Glutamate, glutamine, aspartate, and asparagine are the most abundant, or the most important, amino acids in the organs of germinating lupin seeds. Asparagine in particular is accumulated at a very high level during germination, reaching up to 30% of dry matter [4]. Such high accumulation of this amino acid during lupin seed germination is a result of intense interconversions of other amino acids, for example, through the glutamine synthetase/glutamate synthase cycle (GS/GOGAT cycle) or by the action of glutamate dehydrogenase (GDH). During amino acid metabolism, toxic ammonia is generated and asparagine is accumulated to utilize this adverse by-product of amino acid interconversions [4,5,6] (Figure 1). Amino acids in germinating lupin seeds are not only used for biosynthesis and anabolic reactions in general, but primarily they are the main respiratory substrates, allowing seedling growth and development [2,4,7,8].

Storage lipids are accumulated inside lipid droplets (also called oil bodies, oleosomes, or spherosomes) that are deposited in the cytoplasm [13,14,15] (Figure 1). During seed germination lipases release fatty acids from triacylglycerols. Fatty acids are translocated to the peroxisome (also called the glyoxysome in germinating seeds and vegetative storage tissues or organs), where they undergo β-oxidation. The next step of lipid breakdown is the glyoxylate cycle, which operates partially in the peroxisome and partially in the cytoplasm [2,16]. Succinate, a product of the glyoxylate cycle, is translocated from the peroxisome to the mitochondrion, where it fuels the TCA metabolite pool, and it may be used as a respiratory substrate or, after conversion to oxoacetate, may be translocated back to the cytoplasm. Here, it may become a substrate for phosphoenolpyruvate carboxykinase (PEPCK) and through gluconeogenesis, glucose, and then sucrose, are synthesized [2,17] (Figure 1). Sucrose is the main carbon transport form in plants and the above-described conversion of storage lipid to sugar is typical, occurring in germinating seeds, allowing the growth and development of the seedling. Nevertheless, during lupin seed germination, strong interconnections between storage protein and lipid mobilization occur (Figure 1). We previously found that, during lupin seed germination, some pools of lipid-derived carbon skeletons may be subtracted from the typical, above-described pathway of storage lipid conversion into glucose and sucrose, and can instead be directly used in amino acid metabolism [9]. It is probable that metabolites of the glyoxylate cycle may be such a nodal point of carbon flow from lipids to amino acid synthesis. We described four alternative pathways of carbon skeleton flow from lipids to amino acids in germinating lupin seeds [10].

Our previous experiments on embryonic axes, isolated from lupin seeds and cultured for 96 h in vitro on a mineral liquid medium [18] without sucrose (-S), showed a considerable decrease in soluble sugars content in comparison to axes grown on a medium with 60 mM sucrose (+S) [12], and several symptoms of sugar depletion or starvation occurred, like significantly lower fresh and dry matter content [8,19], a complete degradation of storage protein depositions [11,19] and lack of starch granules [8,11,12], a decrease in soluble protein content [19], an increase in activity of enzymes involved in catabolic processes, for example, proteases degrading storage protein [20], a decrease in respiration efficiency [7,11,21] or a huge increase in cell vacuolization [7,8,11,22,23]. During these studies, we observed that the main storage compound in lupin seeds, i.e., storage protein, was successively used, and the breakdown of this storage compound was enhanced by carbon (sugar) starvation [8,11,19,20]. However, a clear disruption of lipid mobilization in lupin sugar-starved isolated embryonic axes was also observed. The starved axes were characterized by lower fresh and dry mass, and shorter length in comparison to sucrose-fed axes [8,19]. Interestingly, the sugar-starved lupin isolated embryonic axes contained significantly higher amounts of lipid than the sugar-fed ones [8,24]. Such an observation was in contrast to the knowledge about the mobilization of storage compounds during seed germination and the knowledge about the effect of carbon starvation in plant cells, tissues, and organs because it was unambiguously demonstrated that the mobilization of storage compounds during seed germination is enhanced under starvation. It allows the supply of respiratory substrates, the maintenance of cell respiration, and survival under adverse nutrient conditions [23]. Nevertheless, a higher amount of lipid was observed in lupin sugar-starved isolated embryonic axes than in sucrose-fed axes [8,24]. In addition, at the same time, advanced autophagy was also observed in the sugar-starved axes. It was reflected in increased cell vacuolization [8,11] and a decreased level of phosphatidylcholine [24], a metabolic marker of autophagy [25,26]. We also discovered that, in sugar-starved embryonic axes, autophagy may be retarded by exogenous asparagine [8], a central amino acid in the metabolism of germinating lupin seeds [4]. We observed that asparagine significantly slowed down the decomposition of autophagic bodies, one of the final stages of autophagy that takes place in the vacuole. The degradation of autophagic bodies is a rapid process caused by vacuolar lytic enzymes [27]. An observation of the content of the autophagic body without using autophagy mutants or applying autophagy inhibitors is almost impossible [28]. However, by the action of asparagine, we were able to observe the content of autophagic bodies in cells of lupin sugar-starved embryonic axes. Among others, we recognized structures that may be peroxisomes [8]. Taking together the higher content of total lipid in the sugar-starved axes and the possibility of peroxisome degradation through autophagy, we formulated the hypothesis that autophagy causes disturbances in the action of the enzymatic apparatus responsible for lipid breakdown. More precisely, we postulate that, under sugar starvation conditions, pexophagy may occur, i.e., the selective, autophagic degradation of peroxisomes—key organelles involved in lipid mobilization during seed germination [29]. To confirm such a supposition, we performed broad research on different levels. We conducted observations of cell ultrastructure, measured the content of lipids, and assayed enzyme activity. We also performed wide-scale proteomic and transcriptomic analyses. Immunodetection of Pex14p, a peroxisomal marker, in the presence of concanamycin A, an inhibitor of autophagy, was also done. It should be emphasized that we used in the research the embryonic axes of two lupin species that differ in lipid and protein content in seeds, and we also took into consideration the role of asparagine, both during lipid breakdown and during autophagy in germinating lupin seeds.

## 2. Results

Sugar starvation (-S) remarkably enhances autophagy in cells of lupin isolated embryonic axes cultured in vitro for 96 h. One piece of evidence of advanced autophagy in cells of 96 h sugar-starved axes is a clear increase in the vacuolization of root meristematic zone cells in comparison to sucrose-fed axes (+S) (Figure 2). Also, the decrease in the content of phosphatidylcholine and the simultaneous increase in the content of phosphocholine (a product of phosphatidylcholine degradation) can be considered as a metabolic marker of plant autophagy [26]. We detected such changes in the content of these two phospholipids in the cells of sugar-starved axes in comparison to sucrose-fed axes of both lupin species (Figure 3b). Also, we detected the same relation in sugar-starved axes simultaneously fed with asparagine (-S+Asn) compared to the sucrose- and asparagine-fed axes (+S+Asn) (Figure 3b). The third evidence of enhancing autophagy under sugar starvation conditions in white and Andean lupin embryonic axes is a significant increase in the mRNA levels for *ATG8C* in comparison to +S axes (Figure 8; changes in mRNA levels caused by trophic conditions will be described in detail in the subsequent part of this section). Taking together these three observations, we conclude that, in cells of white and Andean lupin, isolated embryonic axes grown in vitro for 96 h, under sugar starvation conditions (-S), advanced autophagy occurs. Our observations of cell ultrastructure also revealed another interesting and intriguing picture that was a result of asparagine action in the sugar-starved (-S+Asn) axes. Namely, in the enlarged vacuoles of root meristematic zone cells, we detected numerous autophagic bodies whose decomposition was remarkably slowed down by asparagine (Figure 2). Pictures obtained from the observation of Andean lupin cells were more spectacular compared to the white lupin ones, but we can conclude that asparagine retards the degradation of autophagic bodies in the sugar-starved axes of both investigated lupin species.

We observed that, under sugar starvation conditions (-S), the degradation of the storage protein in lupin embryonic axes was significantly intensified because we did not observe any protein deposits inside vacuoles of sugar-starved axes in 96 h, while they were visible in sucrose-fed axes (+S), especially in Andean lupin axes additionally fed with asparagine (+S+Asn) (Figure 2). However, we observed completely different relationships concerning the content of total lipid. Namely, the degradation of lipid was more intense in the sucrose-fed axes (+S) than in the sugar-starved axes (-S) of both lupin species. We found that a clear difference in total lipid content between the +S and -S axes of both lupins appears already within 24 h (Figure 3a). Especially in white lupin, this difference was enhanced in the subsequent hours, but finally, it reached a statistically significant level at 96 h in the axes of both lupin species. The highest content of total lipid at 96 h was detected in the embryonic axes of both lupin species which were not fed with sucrose but were fed with asparagine (-S+Asn) (Figure 3a). A slowing in lipid degradation under sugar starvation conditions was also observed at the ultrastructural level, seeing numerous lipid droplets in cells of -S axes but almost no lipid droplets in the +S axes of both lupin species (Figure 2). In particular in the axes of Andean lupin, which accumulates remarkably more lipid in seeds than white lupin, we observed numerous lipid droplets in the cytoplasm, close to the tonoplast (-S+Asn in Figure 2).

Investigating the activity of selected enzymes involved in lipid breakdown during lupin seed germination (Figure 1), we identified a differential pattern of changes caused by sugar starvation (Figure 4a). We discovered that the activity of enzymes catalyzing the initial steps of lipid breakdown was increased under sugar starvation conditions in white and Andean lupin embryonic axes, and they were lipolytic activity, acyl-CoA oxidase, and catalase. The activity of enzymes catalyzing further stages of lipid breakdown was decreased or unchanged in sugar-starved axes compared to sucrose-fed axes. They were cytosolic aconitase, isocitrate lyase, cytosolic malate dehydrogenase, and phosphoenolpyruvate carboxykinase (Figure 4a). Based on such results, we can conclude that the key organelle in such a heterogeneous pattern of enzyme activity is the peroxisome. The activity of enzymes catalyzing reactions in lipid breakdown to the β-oxidation of fatty acids (peroxisome) is increased under sugar starvation conditions, but the activity of enzymes involved in the further stages, i.e., from the glyoxylate cycle, have lower activity in the sugar-starved axes. The effect of asparagine on enzyme activity was also differential for enzymes operating at the beginning and during the later steps of lipid breakdown, and it was opposite to that described above. The activity of the initial enzymes (up to fatty acid β-oxidation) was decreased by asparagine, but the activity of the latest (from the glyoxylate cycle) was elevated, and this relationship was observed both in the sucrose-fed and sucrose-starved axes of both lupin species. The exception was cytoplasmic aconitase (an enzyme of the glyoxylate cycle), whose activity was also decreased by asparagine in the sucrose-starved (-S+Asn) axes of both lupin species (Figure 4a). The effect of sugar starvation on the activity of enzymes involved in amino acid metabolism (NADP-dependent isocitrate dehydrogenase, glutamine synthetase, and asparagine synthetase) was stimulating, but asparagine decreased the activity of these enzymes. The exception was the increased activity of isocitrate dehydrogenase in white and Andean lupin +S+Asn axes in comparison to the +S axes (Figure 4b).

The next step in our research was to determine the nutritionally dependent quantitative changes in enzyme proteins involved in lipid breakdown during lupin seed germination. To achieve this goal we performed iTRAQ-based proteomics analysis. The full iTRAQ data for white and Andean lupin embryonic axes, including the results of statistical analysis for three independent experiments, are presented in Appendix A. From these full iTRAQ data, we selected only those enzymes of lipid catabolism which achieved statistical significance (*q*-value below 0.05, orange filling of a cell in Appendix A) in the protein level changes, and the results of this selection for white lupin embryonic axes are presented in Table 1, and for Andean lupin axes in Table 2. Comparing sucrose-fed (+S) to sucrose-starved (-S) white lupin embryonic axes, we found an increased level of long-chain acyl-CoA synthetase 2-like in the +S axes (Table 1). Proteins of enzymes catalyzing further reactions of lipid breakdown, i.e., reactions of fatty acid β-oxidation and the glyoxylate cycle, represented a lower level in the +S axes in comparison to -S axes of white lupin. They were acyl-CoA oxidase 4, acetate/butyrate-CoA ligase AAE7, Δ^3,5^,Δ^2,4^-dienoyl-CoA isomerase, enoyl-CoA hydratase, 3-ketoacyl-CoA thiolase 2, malate dehydrogenase, and malate synthase. All of these enzymes are located in the peroxisome (glyoxysome) (Table 1). The next step of lipid degradation involves some of the TCA cycle reactions occurring in the mitochondrion (Figure 1). iTRAQ analysis allowed us to identify the increased content of isocitrate dehydrogenase [NAD] catalytic subunit 5 and succinate-CoA ligase [ADP-forming] subunits α and β in +S compared to -S axes (Table 1). The final stage of lipid breakdown is gluconeogenesis, which is initiated by phosphoenolpyruvate carboxykinase (Figure 1). For this enzyme, we detected a lower content of protein in the sucrose-fed (+S) in comparison to sucrose-starved (-S) axes (Table 1). We did not detect statistically significant changes caused by asparagine in the protein level for any of the analyzed enzymes in sucrose-fed axes (the pair +S/+S+Asn in Table 1), while in sucrose-starved axes we identified a statistically significant decrease in the content of malate synthase and isocitrate lyase (the pair -S/-S+Asn in Table 1). Both of these enzymes are involved in the glyoxylate cycle and are the hallmark of peroxisomes [2,16,17]. For Andean lupin embryonic axes, four enzymes had lower content in the +S axes in comparison to the -S axes. They were the peroxisomal enzymes—acyl-coenzyme A oxidase 3, citrate synthase (glyoxysomal-like), and isocitrate lyase 2—as well as one cytoplasmic enzyme—phosphoenolpyruvate carboxykinase [ATP] 1-like (Table 2). Similarly as in white lupin embryonic axes, no statistically significant effect of asparagine on investigated enzyme protein levels was detected in the Andean sucrose-fed axes (the pair +S/+S+Asn in Table 2). In sugar-starved and asparagine-fed (-S+Asn) Andean lupin axes, the contents of the probable enoyl-CoA hydratase, catalase isozyme 1-like, and peroxisomal malate dehydrogenase were increased. In contrast, citrate synthase glyoxysomal-like and aconitate hydratase 1 and 1-like were decreased by asparagine in the sugar-starved axes (the pair -S/-S+Asn in Table 2).

Besides the proteomics analyses, we also performed sequencing of the transcriptome (next-generation sequencing) of white and Andean lupin embryonic axes. Because lupins generate protein-storing seeds, in which the metabolism is based mainly on storage proteins, and amino acids are one of the basic respiratory substrates of germinating seed organs [4], we first checked how the trophic conditions of in vitro culture affect the level of transcripts of genes encoding enzyme proteins involved in amino acid metabolism. Based on data from the literature [4,11,31] and the KEGG database dedicated to the narrow-leaved blue lupin (*Lupinus angustifolius*) genome (https://www.genome.jp/kegg-bin/show_organism?menu_type=pathway_maps&org=lang; accessed on 16 June 2022), we selected records for genes of the central amino acid metabolism in lupins. Heatmaps created from such selected records showed a clear decrease in transcript levels of the majority of genes of the amino acid metabolism in the -S axes compared to +S axes of both investigated lupin species (Figure 5a,b). The feeding with asparagine of the sugar-starved (-S+Asn) embryonic axes of both investigated lupin species maintained or even enhanced the lower transcript level of several genes of amino acid metabolism (Figure 5a,b).

We also found a clear decrease in the level of the majority of transcripts of genes encoding enzymes involved in lipid breakdown in the -S axes in comparison to the +S axes (Figure 6a,b). Asparagine caused an additional decrease in the transcript levels for many analyzed genes in the sucrose-starved (-S+Asn) axes of both lupin species, and similarly as in the amino acid metabolism, asparagine caused a clear increase in transcript levels for many genes in the sucrose-fed axes (+S+Asn), especially in the axes of Andean lupin (Figure 6b). It should be added here that the lowest levels of transcripts of genes related to lipid breakdown were observed in the same axes in which the content of total lipid was the highest at 96 h of in vitro culture among all studied trophic conditions, i.e., in -S+Asn axes (Figure 3a).

We obtained the most unambiguous transcriptomics results for changes in transcript levels for genes encoding proteins responsible for pexophagy, i.e., autophagic degradation of peroxisomes. Based on data from the literature [29,32,33] we prepared a list of genes including *Atg* (autophagy-related) genes, genes encoding pexophagy-related scaffold proteins, as well as genes encoding peroxisomal proteins whose involvement in pexophagy has already been confirmed in yeasts and other plant species. We have observed that, under sugar starvation conditions (-S), almost all analyzed genes represented remarkably elevated levels of transcripts in comparison to sucrose-fed (+S) axes of Andean lupin (Figure 7b). We observed almost the same picture of changes in the axes of white lupin, but for a few genes, the transcript level was much lower in the -S axes than in the +S axes (Figure 7a). The feeding with asparagine of sucrose-starved (-S+Asn) axes caused a decrease in transcript levels for several genes, and this decrease was much more pronounced in white lupin axes (Figure 7a) than in axes of Andean lupin (Figure 7b). Nevertheless, in general, the lowest transcript levels of pexophagy-related genes were observed in the sucrose- and asparagine-fed (+S+Asn) axes of both lupin species, and the asparagine effect was also much clearer in white lupin axes (Figure 7a) than in the Andean lupin ones, in which even solely sucrose (+S) caused low transcript levels for the majority of analyzed genes (Figure 7b). It is worth adding here that the lowest transcript level of the majority of pexophagy-related genes was observed in the +S+Asn axes, i.e., in axes in which the ultrastructural symptoms of autophagy were not observed and not-fully-degraded protein deposits were still visible in vacuoles (Figure 2), and the degradation of lipid was highly effective, as was reflected in low total lipid content in 96 h of the in vitro culture (Figure 3a). Contrary to this, the most elevated transcript levels of pexophagy-related genes were found in the -S axes, i.e., in axes in which huge cell vacuolization (Figure 2) and changes in metabolic markers of advanced autophagy were identified (low content of phosphatidylcholine with simultaneous high content of phosphocholine) (Figure 3b), and lipid breakdown was significantly restricted within 96 h of the in vitro culture (Figure 3a).

For some genes representing all analyzed aspects of our research at the transcriptomic level (Figure 5a,b, Figure 6a,b and Figure 7a,b, we performed RT-qPCR analyses. This technique allowed us to identify a clear increase in the transcript levels for nearly all investigated genes (except for glutamate synthase in white lupin axes) in the -S axes in comparison to the +S axes of both lupin species (Figure 8). Asparagine under sugar starvation conditions (-S+Asn) decreased the transcript levels of all investigated genes in both lupin species. Asparagine also caused the same effect in the sucrose-fed (+S+Asn) axes for the majority of investigated genes, except for catalase isoenzyme 1 in white lupin, and phosphoenolpyruvate carboxykinase [ATP] 1-like in white and Andean lupin axes, whose transcript levels were elevated by this amino acid in comparison to the +S axes (Figure 8). Special attention we focused on *ATG8C* whose transcript level was significantly increased in the -S axes in comparison to the +S axes, and asparagine significantly decreased the transcript level both in the sucrose-fed (+S+Asn) and sucrose-starved (-S+Asn) embryonic axes of both lupin species (Figure 8).

The final step in our research was the determination of the changes in the quantity of Pex14p, which is a peroxisome marker protein. Pex14 is a peroxisome membrane protein which is one of the proteins involved in pexophagy in yeasts [29,36,37]. Our western blot analysis showed a statistically significant decrease in the Pex14p content in the -S axes in comparison to the +S axes of both lupin species (Figure 9a,b). Asparagine in white lupin axes significantly increased the content of Pex14p both in the sucrose-fed (+S+Asn) and sugar-starved (-S+Asn) axes while, in axes of Andean lupin, such an effect was observed only in the sucrose-fed (+S+Asn) axes. During this experiment, we additionally applied concanamycin A—a commonly used inhibitor of autophagy [38]. Using concanamycin A, we observed a clear and statistically significant increase in Pex14p content in both the +S and -S axes of both lupin species (Figure 9a,b). The effect of concanamycin A in axes fed with asparagine was opposite in the +S+Asn and -S+Asn axes. Namely, the content of Pex14p was decreased by the inhibitor in the +S+Asn axes while, in the -S+Asn axes of both lupin species, it was significantly increased (Figure 9a,b).

## 3. Discussion

The mobilization of storage compounds during seed germination is highly regulated. It undergoes metabolic and hormonal control [39] allowing a stable supply of respiratory and anabolic substrates, under both normal and adverse environmental conditions [23]. The optimal sugar level is essential for proper seedling growth and development [40]. The mobilization of storage compounds is enhanced under sugar starvation conditions, and it was already confirmed in storage tissues and organs, including seeds of many plant species [23]. Such intensification in the breakdown of storage compounds under carbon as well as nitrogen deficiency in the sink tissues allows, first of all, the supply of respiratory substrates and hence survival under adverse conditions. In germinating seeds, nutrient deficiency or starvation conditions may appear, for example, as a result of too-deep sowing or damage to the junction between cotyledons and embryonic axes. In lupins, the main storage compound in seeds is protein, constituting up to 50% of seed dry matter [2]. We reported previously that under sugar starvation conditions the protein deposits inside vacuoles were degraded intensively [8,11,19,20,23]. Simultaneously, under sugar starvation conditions, the activity of proteolytic enzymes [20] and the activity of enzymes involved in amino acid metabolism were also enhanced [19,31]. Surprisingly, we observed a completely different picture of the changes caused by sugar deficiency for lipid mobilization during lupin seed germination. This storage compound is accumulated in lupin seeds at a remarkably lower level than storage protein and, depending on the lupin species, it reaches from a few to about 20% of seed dry matter [2]. We have already reported that the total lipid level was significantly higher in the 96 h sugar-starved (-S) lupin embryonic axes than in the axes fed with sucrose (+S) [8,11,24]. We also observed the same relations during this research, but now, we also found that this difference in the content of total lipid between the -S and the +S axes appeared already during the first hours of in vitro culture and reached the highest values at 96 h (Figure 3a). Such a result is very difficult to interpret because it could be expected that, similarly as it was observed in protein mobilization, the breakdown of lipid also will be enhanced under sugar deficiency conditions. Moreover, it is quite difficult to explain the patchy pattern of changes in the activity of enzymes involved in lipid breakdown because we noted that lipolytic activity, as well as the activity of acyl-CoA oxidase and catalase, i.e., enzymes involved in the initial stages of lipid breakdown, were significantly enhanced in the -S axes (Figure 4a) but the total lipid content in these axes was higher (Figure 3a) than in the +S axes of both investigated lupin species. In contrast to enzymes from the beginning of the lipid catabolism pathway, the activity of enzymes involved in further steps of this process, such as isocitrate lyase, cytosolic and mitochondrial aconitase, cytosolic malate dehydrogenase (only in Andean lupin embryonic axes) and phosphoenolpyruvate carboxykinase, was lower under sugar starvation conditions (Figure 4a). We additionally discovered that asparagine action is opposite to the activity of enzymes from the initial and the further stages of the lipid catabolism pathway. The activity of enzymes operating at the initial stages was decreased, while the activity of enzymes from the latest steps of lipid breakdown was increased by this key amino acid for germinating lupin seeds (Figure 4a). For some of the enzymes whose activity we assayed, we also identified proteins in the iTRAQ data. We found that, for white lupin acyl-CoA oxidase (Table 1) as well as for Andean lupin lipase, catalase, and phosphoenolpyruvate carboxykinase (Table 2), the changes in the activity agreed with the changes in the content of enzyme proteins caused by the trophic conditions of the in vitro culture, so we can conclude that the changes in the activity of the above-mentioned enzymes may be a result of the changes in the content of enzyme proteins. But the most important factor is that the ‘switch’ in the regulation of enzyme activity takes place at the stages of lipid degradation occurring in the peroxisome (fatty acid β-oxidation and part of the glyoxylate cycle) (Figure 1).

Under sugar starvation conditions in cells of white and Andean lupin embryonic axes, enhanced autophagy was observed. This was reflected in the significantly elevated vacuolization of the root meristematic zone cells (Figure 2). Also, a significantly decreased content of phosphatidylcholine with, simultaneously, a huge increase in the level of phosphocholine (a product of phosphatidylcholine degradation) in the -S axes compared to the +S axes of both lupin species was detected (Figure 3b). Such mutual change in the content of the above-mentioned phospholipids can be treated as a metabolic marker of advanced autophagy in plant cells during which the degradation of the plasma membrane and its structural components occurs [26]. Additional evidence of advanced autophagy in -S axes can be found in the significantly elevated levels of *ATG8C* gene transcripts in comparison to +S axes (Figure 8). Analyzing the cell ultrastructure, we discovered that asparagine significantly slowed down the degradation of autophagic bodies inside the vacuoles of cells of the sugar-starved (-S+Asn) lupin embryonic axes (Figure 2 and Borek and coworkers [8]). This allowed the observation of the cargo inside the not-fully-degraded autophagic bodies. In this way, we previously found, inside autophagic bodies, organelles that can be identified as peroxisomes [8]. So, if peroxisomes can be degraded during autophagy (pexophagy) under sugar starvation in lupin embryonic axes, the efficiency of the enzymatic apparatus responsible for lipid breakdown may be remarkably disrupted, and the breakdown of lipid can also be significantly impaired, which would be reflected in a higher level of total lipid. Thus, the enhanced autophagy under sugar starvation conditions can be a reason why the content of total lipid was higher in the -S than in the +S axes (Figure 3a). In the +S axes peroxisomes are not autophagically degraded, the enzymatic apparatus operates without disturbances, and lipid can be used efficiently, supplying respiratory and anabolic substrates and allowing better growth [8] of lupin embryonic axes. To confirm such a conclusion, it was very important to determine the changes in the level of Pex14p—a peroxisome protein marker. We detected statistically significantly lower content of this peroxisomal marker in the -S than in the +S axes of both lupins (Figure 9a,b). Such a result may indirectly indicate the lower number of peroxisomes under sugar starvation conditions. Moreover, the level of Pex14p was higher in axes fed with asparagine (both in +S+Asn and -S+Asn axes of white lupin and +S+Asn axes of Andean lupin) (Figure 9a,b). Such a result is compatible with the slower degradation of autophagic bodies caused by asparagine in lupin embryonic axes (Figure 2 and Borek and coworkers [8]). Such an observation may point to the role of autophagy in peroxisome degradation in lupin embryonic axes. To confirm such supposition we applied concanamycin A (autophagy inhibitor) in our Western blot analyses, and we observed, then, a significant increase in Pex14p content. The increase caused by the autophagy inhibitor in the content of this peroxisomal marker was especially evident in Andean lupin sugar-starved axes (both -S and -S+Asn) (Figure 9a,b). This result is especially important because it is evidence of pexophagy in the sugar-starved lupin embryonic axes. Another crucial result supporting the conclusion that pexophagy occurs in lupin embryonic axes is the clear increase in the transcript level of the majority of genes encoding proteins involved in pexophagy that we observed in the -S axes in comparison to the +S axes of both lupin species (Figure 7a,b). Taking together the results presented in this paper, as well as our previously published results, like (i) the lower content of the peroxisome marker Pex14p and its clear elevation by an autophagy inhibitor (Figure 9a,b) in the -S axes in comparison to the +S axes, (ii) the higher transcript level of pexophagy machinery genes (Figure 7a,b), including *ATG8C* (Figure 8) in the -S axes than in +S axes, as well as (iii) the possible peroxisome localization inside autophagic bodies [8], we can conclude that, under sugar starvation conditions in lupin embryonic axes, autophagic degradation of peroxisomes (pexophagy) occurs and contributes to disruption in lipid breakdown.

It should be emphasized that the research was conducted on the embryonic axes of seeds whose main storage compound is a protein [2]. The metabolism of such seeds is based primarily on amino acids, which are one of the main respiratory and anabolic substrates. Under sugar starvation conditions the protein content in lupin isolated embryonic axes is significantly decreased [19,20,22,23,31]. Due to the deficit of protein and amino acids under sugar starvation conditions, in general, the metabolism of lupin embryonic axes can be slowed down, which is reflected in a decreased respiratory rate [7,11,21] and slower growth of lupin embryonic axes [8,19]. Exogenously added asparagine, one of the most important amino acids for growing lupin embryonic axes [2,4,8], undoubtedly supplemented the deficit of respiratory and anabolic substrates, allowing better growth under such circumstances [8]. Asparagine in plants is converted into aspartate and ammonia by the action of asparaginase [41]. Ammonia is utilized in the GS/GOGAT cycle and can be incorporated into amides and amino acids other than asparagine [42]. Aspartate, on the other hand, after transamination is converted into oxaloacetate [43], an intermediate of the TCA cycle, thereby allowing respiration to be sustained under sugar starvation conditions. Thus, the global amino acid metabolism in such adverse conditions can be slowed down, and it can be restricted, first of all, to necessary reactions sustaining respiration and allowing survival. By contrast, in the sucrose-fed embryonic axes (+S+Asn), especially in Andean lupin axes, we detected the stimulatory effect of asparagine on the level of the transcripts of the majority of genes related to the central amino acid metabolism (Figure 5a,b). It should be noted here that embryonic axes of both lupin species feeding simultaneously with sucrose and asparagine (+S+Asn) grow the most efficiently during 96 h of in vitro culture, which was reflected in the highest fresh and dry matter content [8] as well as in the highest utilization of storage protein [8] among all investigated trophic conditions. So, the enhanced expression of amino acid metabolism genes (Figure 5a,b) under such favorable trophic conditions is not surprising. Nevertheless, under nutrient depletion or starvation conditions, the increased demand for respiratory substrates occurs [44] and the acquisition of such substrates is remarkably easier from the degradation of storage protein than from storage lipid. If the source of respiratory and anabolic substrates is storage protein, amino acids are released from albumins and globulins (in lupins they are mainly globulins, constituting up to 80% of all storage proteins [1]) by the action of endo- and exopeptidases [20], and such amino acids are almost ready to supply the respiration. For the conversion of amino acids between themselves, only deamination and transamination reactions suffice, and then amino acids may directly supply the TCA cycle metabolite pool [4,11] (Figure 1). Amino acids can also be a form of carbon and nitrogen transport in plants [45]. If, on the other hand, the source of the respiratory and anabolic substrates is the storage lipid, considerably more conversion steps are needed. For the full conversion of triacylglycerols, the main compound of the storage lipid, into glucose or sucrose, about thirty reactions are needed [2]. Such full conversion occurs under normal growth and development conditions, and sucrose is the main carbon transport form in plants [45,46], including lupins [2]. Under starvation conditions, in germinating lupin seeds, storage lipid does not have to be converted to the transport sugar but may also be a source of respiratory substrates [9,11]. Nevertheless, the conversion of triacylglycerols to metabolites of the TCA cycle needs considerably more reactions than the release of amino acids from the storage proteins and their use as respiratory substrates (Figure 1). Due to such a multi-stage catabolic pathway, the acquisition of respiratory substrates from lipids under sugar starvation conditions in cells of lupin embryonic axes probably is not favorable and it even seems to be ‘risky’ because the demand for amino acids for sustaining respiration is certainly higher under such circumstances in protein seeds such as lupin seeds. To sustain respiration, amino acids can be acquired from various sources. One of them can be cytosolic proteins, including enzyme proteins involved in lipid breakdown. Apart from cytosolic enzymes, also whole organelles can be degraded to obtain the respiratory substrates under adverse nutrient conditions. Thus, peroxisomes, which harbor many enzymes of a variety of processes occurring in a plant cell [33,34,47,48,49,50], are rich in proteins and can be a good source of amino acids as well, especially in plant tissues and organs whose metabolism is based on amino acids. If so, the degradation of peroxisomes through autophagy, even though the storage lipid is not yet fully used, is justified or even necessary to allow survival under starvation conditions. 

Our ultrastructural observations indicated that lipid droplets were remarkably less numerous but larger (Figure 2) than usually can be observed in lupin tissues [8,11,12,19,22]. Such enlargement of lipid droplets may suggest a disruption in the level of proteins specific for lipid droplets (oleosins, caleosins, and steroleosins). These proteins are anchored in the lipid droplet’s monolayer membrane [14,51,52,53]. Oleosins in particular are important for stabilizing the structure of these organelles in the cytoplasm and, due to their negatively charged domains [53,54], keep oleosomes from fusing [13] despite rapid tissue dehydration and rehydration during seed desiccation and imbibition [51]. Studies performed on *Arabidopsis* and maize showed that oleosin depletion leads to the appearance of unusually large and structurally abnormal lipid droplets [13,53]. The small size of lipid droplets allows, among other things, the effective breakdown of lipids during seed germination. In other words, the small size of lipid droplets enlarges their surface and surface-anchored lipolytic enzymes (for example lipases) can effectively degrade storage acylglycerols. The enlarged lipid droplets (Figure 2) and higher total lipid content (Figure 3a) that we observed in sugar-starved embryonic axes (-S and -S+Asn), in comparison to sucrose-fed axes (+S+Asn), proved the disruption in lipid breakdown. Moreover, our ultrastructure observations showed that lipophagy can occur in cells of lupin embryonic axes under sugar starvation conditions. Lipophagy is an autophagic degradation of whole lipid droplets [14,55]. It was especially clearly visible in cells of Andean lupin embryonic axes (Figure 2), i.e., the lupin species accumulating remarkably more lipid in seeds than white lupin [2]. The statement that lipophagy may occur in cells of the sugar-starved lupin embryonic axes is supported by the unambiguous identification of lipid droplets inside not-fully-degraded autophagic bodies (-S+Asn in Figure 2). However, lipophagy does not seem to be a preferential way of degradation of the remaining lipid in lupin embryonic axes because only a few lipid droplets were visible inside non-degraded autophagic bodies and most of them stayed in the cytoplasm, showed a tendency to merge and increase in size, and accumulated close to the tonoplast. They seemed to be ‘pressed’ into the vacuole, but they were not very numerous in the autophagic bodies (Figure 2). So, it can be concluded that lipophagy did not occur very intensively in the cells of the sugar-starved Andean lupin embryonic axes. Perhaps lipophagy would be more important with the extension of the starvation period and the deterioration of nutrient conditions. Our research, including ultrastructure observations, was performed on embryonic axes cultured in vitro for up to 96 h, and confirming the above supposition would require further research with an extension of the starvation period to more than 96 h.

## 4. Materials and Methods

### 4.1. Plant Material

Seeds of white lupin (*Lupinus albus* L.; up to 14% total lipid content in seed dry matter) and Andean lupin (*Lupinus mutabilis* Sweet; about 20% total lipid content in seed dry matter) were surface-sterilized in 0.02% HgCl_2_ for 15 and 20 min, respectively, washed five times with sterile water and allowed to imbibe in the dark for 24 h at 25 °C. Embryonic axes isolated from imbibed seeds were placed on sterilized filter paper (Whatman no. 3) in sterile tubes on Heller medium [18] with (+S) and without (-S) 60 mM sucrose. Culture media were also enriched with 35 mM asparagine (+Asn). Asparagine-enriched media were sterilized with 0.22 mm Millipore filters. Isolated embryonic axes were cultured in vitro for 96 h in the dark at 25 °C. The morphology of embryonic axes is presented in Appendix A and detailed physio-morphological parameters were published earlier by Borek and coworkers [8].

### 4.2. Ultrastructure

Ultrastructural observations were conducted in cells of the root meristematic zone of isolated embryonic axes that were cultured in vitro for 96 h. Tissues were prepared according to Borek and coworkers [22]. Root tips (3 mm long) were fixed in a mixture of 4% glutaraldehyde and 4% formaldehyde (1:1, *v*/*v*) in cacodylate buffer (0.05 M, pH 6.8), post-fixed in 1% OsO_4_, and stained in 2% uranyl acetate. Spurr low-viscosity epoxy resin was used. Ultrathin sections (70 nm thickness) were prepared using Ultracut S (Reichert, Depew, NY, USA), stained in 5% uranyl acetate and 0.5% lead citrate, and observed under the transmission electron microscope TEM-1200Ex (JEOL, Tokyo, Japan). To be sure, the cells of the root meristematic zone were observed under a transmission electron microscope, the root tips embedded in resin were successively cross-sectioned into semithin sections (2.5 μm thickness) starting from the tip of the root cap. Under the control of a light microscope, the root meristematic zone was identified, and then ultrathin sections were prepared for this area. A detailed description and graphical presentation of the localization of the area of ultrastructure observations is provided in Appendix A.

### 4.3. Total Lipid and Phospholipids

Lipid and phospholipid content was measured as described by Borek and coworkers [56]. Total lipid was extracted from isolated embryonic axes cultured in vitro using chloroform:methanol (2:1, *v*/*v*) containing 0.05% butylated hydroxytoluene. The amounts of total lipids were determined gravimetrically—an aliquot (500 µL) of the lipid extract was transferred to a previously weighed aluminum weighing bottle (1 cm diameter) and weighed, then dried under a stream of nitrogen gas to evaporate the extraction mixture and reweighed to obtain the dry weight of the extract. 

Phospholipids were assayed in an aliquot of lipid extract separated by one-dimensional TLC in chloroform:methanol:acetic acid:water (85:15:10:3.5, *v*/*v*) using original phospholipid standards. Spots containing phosphatidylcholine, phosphatidylinositol, phosphatidylethanolamine, phosphatidylglycerol, and phosphatidic acid (detected by iodine vapor) were scraped for phosphorus content analysis according to Ames, 1966.

The content of phosphocholine was measured as follows: (i) to 200 μL of the water fraction obtained in the isolation of phospholipids, 100 μL of 0.1 M sodium acetate (pH 5.0), 50 μL of acid phosphatase (40 U/mL), and 150 μL water were added. The mixture was incubated at 37 °C for 16 h. (ii) 600 µL of the reaction mixture—500 µL of 0.1 M HEPES-Na (pH 8.0), 10 µL of 0.1 M 4-amino antipyrine, 10 µL of 0.2 M phenol, 10 µL of choline oxidase (200 U/mL), 10 µl of peroxidase (500 U/mL), and 60 µl of water—was incubated at 37 °C for 20 min. The content was monitored at 500 nm and the sum of choline and phosphocholine was obtained. Choline content was measured without the use of acid phosphatase (point i). To obtain the phosphocholine content, the choline content was subtracted from the sum of choline and phosphocholine.

### 4.4. Enzyme Activity Assays

Lipolytic activity was assayed spectrophotometrically according to the procedure described by Borek and coworkers [22] using *p*-nitrophenol palmitate as the substrate. The activity of catalase (EC 1.11.1.6), using H_2_O_2_ as the substrate, and cytosolic isocitrate dehydrogenase (EC 1.1.1.42), using isocitrate and NADP^+^ as substrates, were assayed according to spectrophotometric methods described in detail by Borek and coworkers [22]. The activity of cytosolic aconitase (EC 4.2.1.3), using sodium LD-isocitrate as the substrate, and isocitrate lyase (EC 4.1.3.1), using sodium isocitrate and reduced glutathione as substrates, were assayed spectrophotometrically as described in detail by Borek and Nuc [57]. The activity of acyl-CoA oxidase (EC 1.3.3.6) was assayed by following H_2_O_2_ formation in a coupled assay which determines H_2_O_2_ in peroxidatic reaction as described by Borek and coworkers [12]. Phosphoenolpyruvate carboxykinase activity (EC 4.1.1.49) was assayed spectrophotometrically in the carboxylation direction in which oxaloacetate formed is reduced to malate by malate dehydrogenase and NADH as described in detail by Borek and coworkers [12]. The activity of cytosolic malate dehydrogenase (EC 1.1.1.37) using oxaloacetate and NADH as substrates was measured spectrophotometrically according to Hipkin and Syrett [58]. Glutamine synthetase (EC 6.3.1.2) was measured colorimetrically as its transferase activity using glutamine as the substrate and ferric chloride as the staining agent according to Hipkin and Syrett [58]. The activity of asparagine synthetase (EC 6.3.5.4) was measured colorimetrically using aspartate and glutamine as substrates and ninhydrin as the staining agent according to Loureiro and coworkers [59]. The quality of cytosolic fractions used for assays of isocitrate dehydrogenase and aconitase activity was evaluated by assaying the activity of one of the mitochondrial markers, i.e., NADH-GDH (EC 1.4.1.2) using 2-oxoglutarate, ammonium chloride, and NADH as substrates following spectrophotometric method described by Borek and coworkers [19]. Protein concentration in enzyme extracts was determined according to Bradford’s method [60], with BSA as a standard.

### 4.5. Proteomics—iTRAQ

Lupin embryonic axes (1 g) were homogenized in liquid nitrogen to obtain a fine powder, which was resuspended in homogenization buffer 0.1 M Tris-HCl, pH 7.5, with 1 mM EDTA, 1 mM DTT, 1% Protease Inhibitor Cocktail for plant cell and tissue extracts (Sigma-Aldrich, St. Louis, MO, USA), 0.1% Triton X-100, and 10% glycerol and incubated for 30 min in an ice bath. After centrifugation for 10 min at 12,000× *g*, the protein fraction was precipitated in 10% trichloroacetic acid (TCA) in acetone at −20 °C overnight followed by three subsequent washing steps in cold acetone and centrifugation [61]. The protein pellet was dried and dissolved in 100 μL of 0.1 M Tris-HCl buffer, pH 7.5, with 8 M urea and 4% CHAPS by vortex mixing for 2 h at room temperature followed by centrifugation for 15 min at 15,000× *g*. Protein concentration was measured using the modified Bradford’s method [60]. The modification was performed using the above-mentioned buffer for the quantification of protein and preparation of the standard curve to eliminate the interference effect [62]. The protein samples were sent to the Laboratory of Mass Spectrometry of the Institute of Biochemistry and Biophysics, Polish Academy of Science in Warsaw, Poland for further analysis.

Protein samples (100 µg) were prepared according to the FASP protocol with minor modifications [63]. Next, peptide concentration was measured and samples were vacuum-dried. Dried peptides were resuspended in 30 µL of 100 mM TEAB buffer and labeled with iTRAQ 8plex (SCIEX, Framingham, MA, USA) tags in 110 μL of isopropanol for 2 h on vortex. The reaction was quenched by the addition of 8 µL of 5% hydroxylamine. The labeling efficiency was checked and the combined iTRAQ sample was desalted using two 30 mg Oasis HLB columns (Waters, Milford, MA, USA). Aliquots were dried and resuspended in 500 µL of 10 mM ammonium hydroxide.

iTRAQ-labeled peptides were fractionated using high-pH reverse-phase chromatography on the XBridge Peptide BEH C18 column (4.6 × 250 mm, 130 Å, 5 µm, Waters). Separation was performed at a 1 mL/min flow rate for 27 min using the Waters Acquity UPLC H-class system. Mobile phases consisted of water (A), acetonitrile (B), and 100 mM ammonium hydroxide solution (C). The percentage of phase C was kept at a constant 10% through the entire gradient. The following gradient was applied: 3 to 5% solvent B for 0.5 min, 5 to 22% for 16.5 min, 22 to 28% for 2 min, 28 to 45% for 1.5 min, 45 to 90% for 0.5 min, 2.5 min isocratic hold at 90%, and final column equilibration at 3% phase B for 3 min. A total of 25 fractions were dried in a Speedvac and reconstituted in 0.1% formic acid in water by 15 min sonication.

Fractions were analyzed using an LC-MS system consisting of a UPLC chromatograph (nanoAcquity, Waters) and a Q Exactive mass spectrometer (Thermo, Waltham, MA, USA). Peptides were trapped on a C18 pre-column (180 µm × 20 mm, Waters) using a 0.1% water solution of FA as a mobile phase and then transferred to a BEH C18 column (75 µm × 250 mm, 1.7 µm, Waters) using an ACN gradient (0–35% ACN in 160 min) in the presence of 0.1% FA at a flow rate of 250 nL per min. Data acquisition was carried out using a data-dependent method with the top 12 precursors selected for MS2 analysis after collisional-induced fragmentation (CID) with an NCE of 27. The mass spectrometry proteomics data for white and Andean lupin embryonic axes have been deposited to the ProteomeXchange Consortium via the PRIDE [64] partner repository with the dataset identifier PXD041380, Project DOI: 10.6019/PXD041380, https://www.ebi.ac.uk/pride/, deposited on 6 April 2023.

The acquired MS/MS data were pre-processed with Mascot Distiller software (v. 2.6, MatrixScience, London, UK) and a search was performed with the Mascot Search Engine (MatrixScience, London, UK, Mascot Server 2.5) against the NCBInr *Lupinus* database (61,158 sequences; 33,890,079 residues). To reduce mass errors, the peptide and fragment mass tolerance settings were established separately for individual LC-MS/MS runs after a measured mass recalibration [30]. The rest of the search parameters were as follows: enzyme, trypsin; missed cleavages, 1; fixed modifications, methylthio (C), iTRAQ8plex (N-term), iTRAQ8plex (K); variable modifications, oxidation (M), iTRAQ8plex (Y); and instrument, HCD. The statistical significance of peptide identifications was estimated using a joined target/decoy database search approach. This procedure provided *q*-value estimates for each peptide spectrum match (PSM) in the dataset. All PSMs with *q*-values above 0.01 were removed from further analysis. A protein was regarded as confidently identified if at least two of its peptides were found. Proteins identified by a subset of peptides from another protein were excluded from the analysis. The mass calibration and data filtering were carried out with MScan software, developed in-house (the Mass Spectrometry Laboratory at the Institute of Biochemistry and Biophysics of Polish Academy of Sciences, https://mslab-ibb.pl/en/science/software). The lists of peptides and proteins, as well as reporter ion intensities, were exported for further analysis. Statistical significance was assessed with the Diffprot in-house software package for statistical significance assessment [30]. Calculated *p*-values were adjusted for multiple testing using a procedure controlling for false discovery rate (FDR). Only proteins with a *q*-value below 0.05 or those present in only one of two compared analytical groups were taken into consideration during further analysis. Complete, processed iTRAQ data for white and Andean lupin embryonic axes, including the results of statistical analysis for three independent experiments, are presented in Appendix A.

### 4.6. Western Blot

Embryonic axes were homogenized on ice in the lysis buffer (7 M urea, 2 M thiourea, 4% CHAPS, 35 mM Tris) in the proportion of 2 mL per 1 g of fresh weight. Protein measurement was performed according to the method described by Bradford [60], using bovine serum albumin (BSA) as the standard. Samples containing 40 µg of total protein were denatured in Laemmli buffer with 25 mM DTT at 96 °C for 3 min and were separated on 12.5% SDS-PAGE [65]. Electrophoresis was carried out using the Mini-PROTEAN Tetra Cell (Bio-Rad Laboratories, Inc., Grand Junction, CO, USA). Separated proteins were transferred to a PVDF membrane (Immobilon-P, pore size of 0.45 µm), using a semi-dry transfer system (Sigma-Aldrich, St. Louis, MO, USA). The transfer was carried out at 1.25 mA for each square cm for 60 min. The blot was blocked with a solution containing 3% BSA in Tris-buffered saline (TBS) overnight with agitation at 4 °C and next incubated with the primary antibody Pex14p (peroxisomal marker; Agrisera, Vannas, Sweden, catalogue no. AS08 372) at a dilution of 1:10,000 in a blocking solution containing 1% BSA in TBS, for 2 h at room temperature, with agitation. The antibody solution was decanted, and the blot was washed four times for 5 min in TBS at room temperature with agitation. The secondary antibody was goat anti-rabbit IgG horseradish peroxidase (HRP) conjugated (Agrisera, Vannas, Sweden) diluted to 1:10,000 in a blocking solution containing 1% BSA in TBS. After 2 h of incubation at room temperature with agitation, the blot was washed five times for 5 min in TBS at room temperature. Bands were digitally registered using the ChemiDoc MP Imaging System (Bio-Rad, Hercules, CA, USA) after 120 s of incubation with the Agrisera ECLSuperBright (Agrisera, Vannas, Sweden). The exposure time was 20 s. The protein extraction and immunoblots were performed in triplicate.

To assess the role of autophagy in the changes in peroxisomal marker Pex14p level, before the western blot analysis, the embryonic axes were treated with concanamycin A (an inhibitor of autophagy [38]). After 72 h of the in vitro culture, for the 24 h before the western blot analysis, axes were transferred to media containing 10 μM concanamycin A dissolved in 0.1 mM DMSO. The control axes were transferred at the same time to media with the addition of the same aliquots of 0.1 mM DMSO without the inhibitor.

### 4.7. Transcriptomics—NGS

Lupin embryonic axes cultured in vitro under the conditions described above were collected, immediately frozen in liquid nitrogen, and stored at −80 °C until RNA isolation. Frozen embryonic axes were powdered in liquid nitrogen and total RNA was isolated from samples of 50 mg using the RNeasy Plant Mini Kit (Qiagen, Hilden, Germany) without any changes to the protocol. The isolated RNA from each sample was analyzed for its quality and quantity. Equal amounts (4 µg) of total RNA from each probe were used for the preparation of cDNA libraries for next-generation sequencing (NGS). cDNA libraries were prepared using TruSeq Stranded mRNA Sample Preparation kit v2 (Illumina RS-122-2101) using all the necessary additional reagents, such as Agencourt AMPure XP beads (Beckman Coulter, Brea, CA, USA), SUPERSCRIPT III (Invitrogen, Carlsbad, CA, USA), and Fast DNA end repair kit (Thermo Fisher Scientific, Waltham, MA, USA). According to the protocol, polyA-containing mRNA molecules were purified using two rounds of purification on polyT oligo-attached magnetic beads and then fragmented. The first and second strands of cDNAs were synthesized, end-repaired, and then adaptors were ligated after adenylation at the 3′ ends. The quantity of indexed libraries was estimated by a Qubit fluorimeter (Thermo Fisher Scientific), mixed in equal amounts, and sent for NGS to Fasteris (Life Science, Genesupport SA, Plan-les-Ouates, Switzerland). Each time before high-throughput NGS, the quality and pooling process was analyzed by MiSeq (preliminary run exploiting 50 bp). Libraries were sequenced by exploiting the 150 bp paired-end protocol. Parameters describing the quality of cDNA libraries and NGS are presented in Appendix A. The in vitro culture of the embryonic axes, RNA isolation, and NGS were performed with three independent replicates. Raw transcriptomics data have been deposited to the SRA database. The BioProject accession number for white lupin: PRJNA953600, https://www.ncbi.nlm.nih.gov/sra/PRJNA953600, deposited on 11 April 2023, and Andean lupin: PRJNA953433, https://www.ncbi.nlm.nih.gov/sra/PRJNA953433, deposited on 9 April 2023.

Sequencing raw data after demultiplexing (splitting of the raw sequencing data into separate files using sample-specific barcodes) were cleaned by eliminating adapter reads, N-base reads, and low-quality reads using the CLC Genomics Workbench trim sequences module (Qiagen version 20). For expressed gene analysis, the expression level of each gene in each library was calculated by quantifying the number of Illumina reads that mapped to each LupAngTanjil_v1.0 reference sequence annotated with cDNAs (number of contigs 57,263) using the CLC Genomics Workbench RNA-Seq Analysis module. The raw gene expression counts were normalized using the RPKM method described by Mortazavi and coworkers [66]. The RPKM value represents the reads per kilobase of the exon model per million mapped reads. Heatmaps from selected records were made by the online software Heatmapper http://www.heatmapper.ca, using ‘normalized means’ from three independent sequencings (Appendix A).

### 4.8. qRT-PCR

To perform qRT-PCR reactions, total RNA was extracted from 200 mg samples of isolated embryonic axes frozen (−80 °C) and powdered in liquid nitrogen using the RNeasy Plant Minikit (Qiagen, Hilden, Germany) according to the supplier’s recommendations. The concentration of each RNA sample was measured using a NanoDrop 2000 spectrophotometer (Thermo Scientific, Waltham, MA, USA). Only the RNA samples with a 260/280 ratio between 1.9 and 2.1 were used for the analysis. The integrity of RNA samples was also assessed by agarose gel electrophoresis and purity was confirmed by PCR using actin-specific primers (Appendix A). Then, 3 μg of total RNA was used for cDNA synthesis with oligo(dT)_20_ (50 μM) primers and the Superscript III Reverse Transcriptase Kit (Invitrogen). qRT-PCR reactions were carried out using a CFX96 Real-Time PCR Detection System (Bio-Rad) and iTaq Universal SYBR Green Supermix (Bio-Rad), and the specific primers for tested genes (Appendix A). Primers were designed on NGS-derived sequences mapped to the reference genome. The comparative C_T_ method for relative quantification was used with actin as an endogenous control. The amount of target, normalized to an endogenous reference and relative to the calibrator (+S trophic variant), is given by 2^−ΔΔCT^ [35]. Genes for qRT-PCR reactions were chosen among genes included in heatmaps (Figure 5a,b, Figure 6a,b and Figure 7a,b).

### 4.9. Statistics

All experiments were performed in three independent replicates. Data presented on graphs are the averages of three replicates ± SD. The results were subjected to ANOVA statistical analysis and Tukey’s HSD multiple-range test using Statistica software Version 13 (TIBCO Software Inc., Palo Alto, CA, USA). iTRAQ and RNA-Seq (NGS) were performed in three independent biological replicates. Statistical analysis of iTRAQ data was performed with Diffprot software [30], implementing a non-parametric test for assessing significance, with correction for multiple testing.

## 5. Conclusions

Data presented in this paper as well as results of our previous research [8,24] show that, in the embryonic axes of white and Andean lupin germinating seeds, disruption of lipid breakdown occurs under sugar starvation conditions. This is reflected in the higher content of total lipid in the sucrose-starved (-S) than in the sucrose-fed (+S) organs. One of the reasons for this disorder can be the autophagic degradation of peroxisomes (pexophagy)—a key organelle in lipid catabolism. Evidence of pexophagy in cells of sugar-starved lupin axes includes the higher transcript level of genes encoding proteins of pexophagy machinery and the lower content of the peroxisome marker Pex14p and its increase caused by an autophagy inhibitor (concanamycin A) in the -S axes in comparison to the +S axes. Also, autophagic degradation of whole lipid droplets (lipophagy) may occur in the sugar-starved lupin embryonic axes, but this process does not seem to have great importance in lipid utilization during adverse nutrient conditions, at least until 96 h of axis growth. We also conclude that, despite the different composition of the storage compounds between seeds of white and Andean lupin, including a clear difference in the total lipid content, the general picture of lipid breakdown during germination is very similar in both of these investigated species. The results of our research also demonstrate that autophagy occurs at the very early stages of plant growth and development, even in the cells of embryonic seed organs, and it is intensified under nutrient deficit or starvation conditions, enabling survival in unfavorable circumstances.

## Figures and Tables

**Figure 1 ijms-24-11773-f001:**
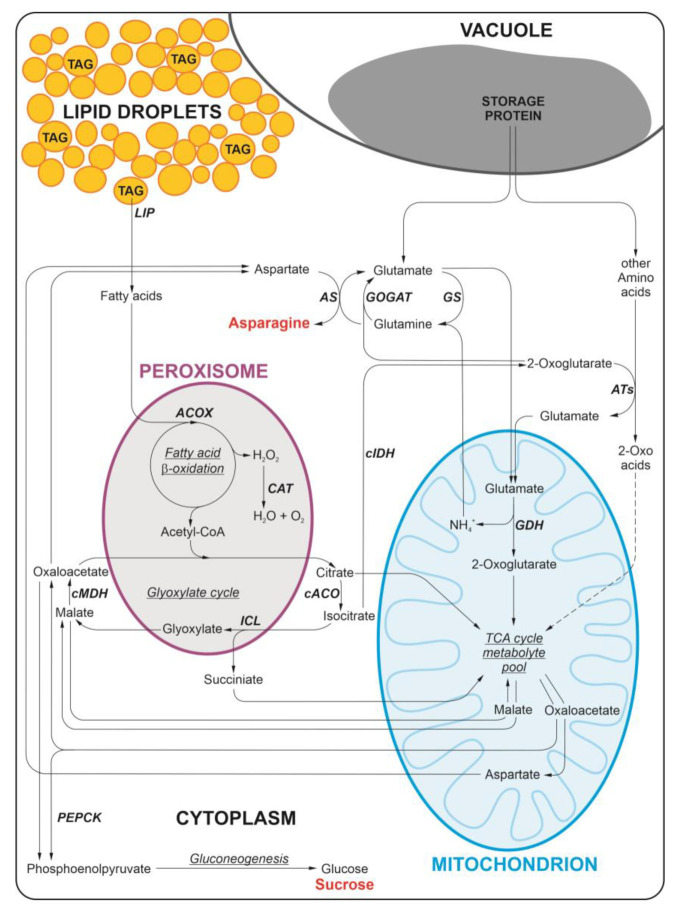
Simplified scheme of interconnections between storage lipid and protein breakdown in germinating lupin seeds. During lipid breakdown, fatty acids released by lipases from triacylglycerols (TAG) deposited in lipid droplets are oxidized in the peroxisome through β-oxidation. The next step of lipid breakdown involves the glyoxylate cycle, which operates partially in the peroxisome (also known as the glyoxysome) and the cytoplasm. The final product of lipid mobilization is sucrose, which is one of the most important carbon transport forms, and it can also be used as a respiratory substrate. However, in germinating lupin seeds, the main respiratory substrates are amino acids. With the intense amino acid interconversions in tissues of lupin germinating seeds, toxic ammonia is generated, and to maintain a low level of this byproduct, asparagine is synthesized in high amounts to utilize ammonia. Some metabolites of the glyoxylate cycle are subtracted from the typical pathway of lipid breakdown and can be used as respiratory substrates as well or be involved in amino acid metabolism. It was evidenced that in germinating lupine seeds, a pool of lipid-derived carbon skeletons is directed for amino acid synthesis, and glutamate and asparagine are among the main targets of carbon flow from lipid to amino acids. Based on data from the literature [2,4,9,10,11,12]. ACOX Acyl-CoA oxidase, AS asparagine synthase (glutamine-hydrolyzing), ATs aminotransferases, cACO aconitase (cytosolic fraction), cIDH isocitrate dehydrogenase (NADP-dependent, cytosolic fraction), cMDH malate dehydrogenase (cytosolic fraction), GDH glutamate dehydrogenase, GOGAT glutamate synthase, GS glutamine synthetase (glutamate-ammonia ligase), ICL isocitrate lyase, CAT catalase, LIP lipase, PEPCK phosphoenolpyruvate carboxykinase, TAG triacylglycerol.

**Figure 2 ijms-24-11773-f002:**
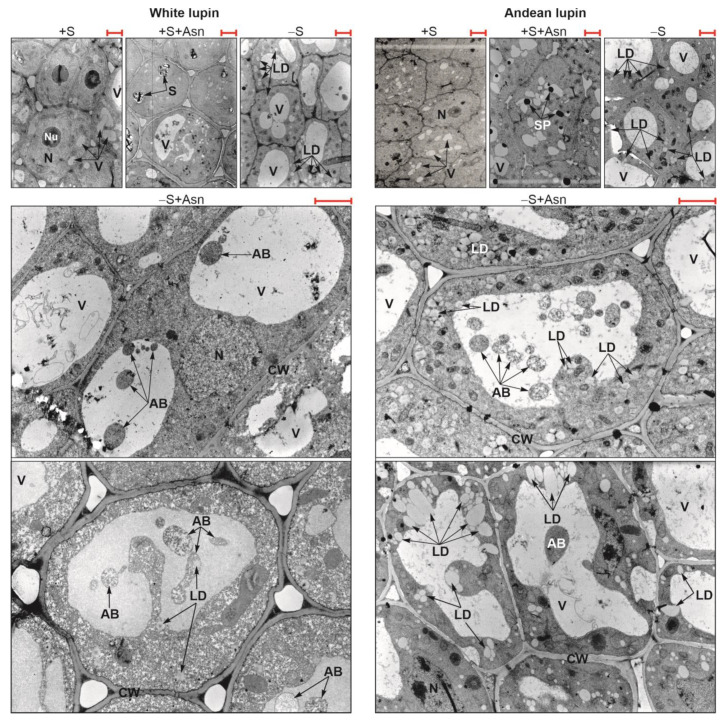
Ultrastructure of root meristematic zone cells of white and Andean lupin isolated embryo axes cultured in vitro for 96 h on a medium with (+S) and without (-S) 60 mM sucrose. Culture media were also enriched with 35 mM asparagine (+Asn). AB—autophagic body, CW—cell wall, LD—lipid droplet, N—nucleus, Nu—nucleolus, S—starch, SP—storage protein, V—vacuole. Scale bars (red; above the micrographs) = 2.0 μm.

**Figure 3 ijms-24-11773-f003:**
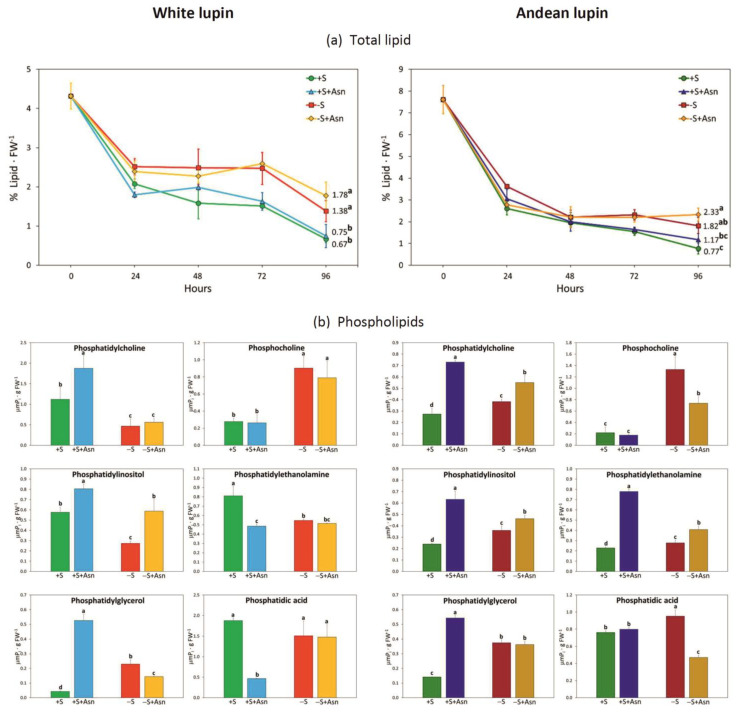
Total lipid content (**a**) and phospholipids (**b**) in white and Andean lupin isolated embryo axes cultured in vitro for 96 h on a medium with (+S) and without (-S) 60 mM sucrose. Culture media were also enriched with 35 mM asparagine (+Asn). Different letters above the error bars (±SD) indicate statistically significant differences at *p* ≤ 0.05 (ANOVA, Tukey’s HSD multiple-range test).

**Figure 4 ijms-24-11773-f004:**
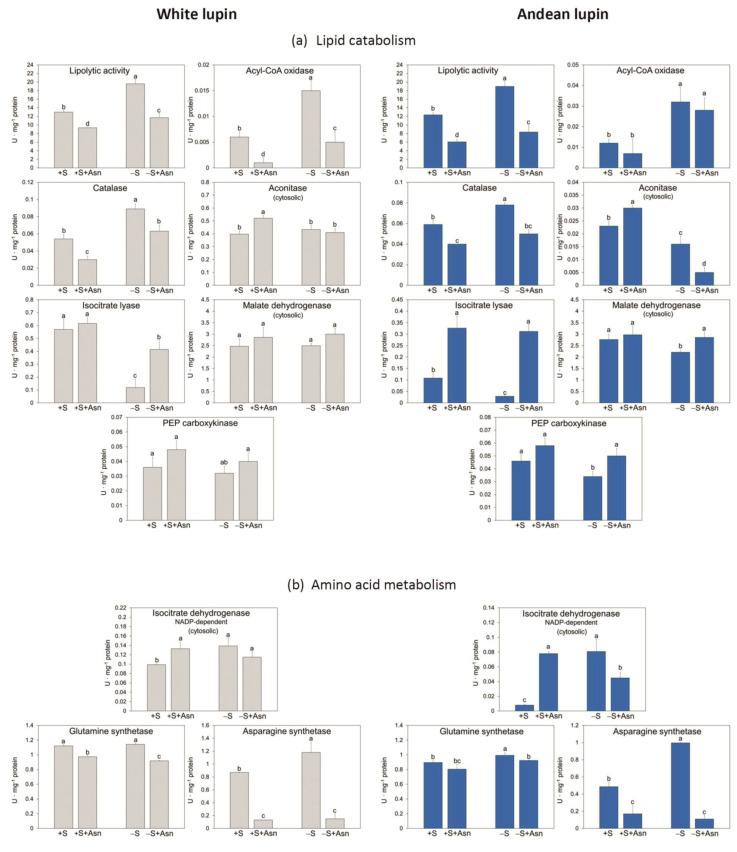
The activity of selected enzymes involved in lipid breakdown (**a**) and amino acid metabolism (**b**) in white and Andean lupin isolated embryo axes cultured in vitro for 96 h on a medium with (+S) and without (-S) 60 mM sucrose. Culture media were also enriched with 35 mM asparagine (+Asn). Different letters above the error bars (±SD) indicate statistically significant differences at *p* ≤ 0.05 (ANOVA, Tukey’s HSD multiple-range test).

**Figure 5 ijms-24-11773-f005:**
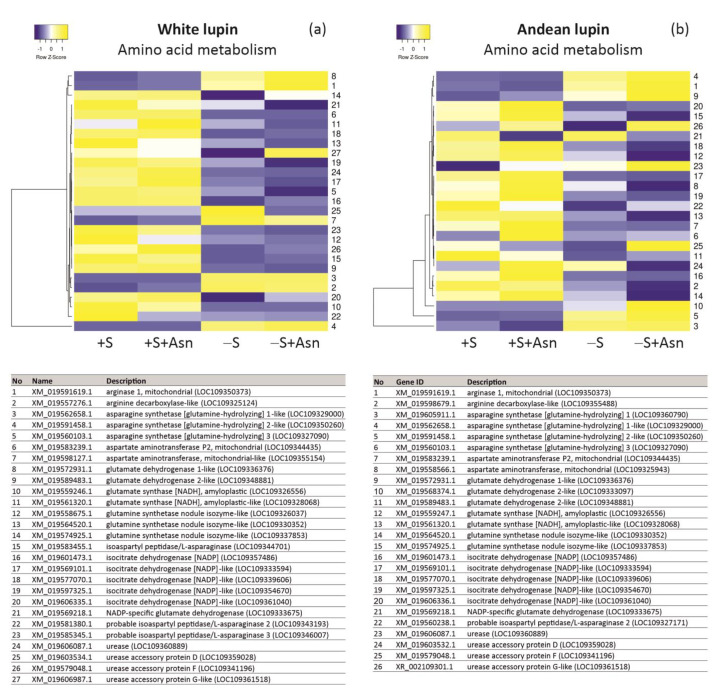
Heatmaps showing the relations in the level of transcripts of genes encoding proteins involved in central amino acid metabolism in white lupin (**a**) and Andean lupin (**b**) isolated embryonic axes cultured in vitro for 96 h on a medium with (+S) and without (-S) 60 mM sucrose. Culture media were enriched with 35 mM asparagine (+Asn). The data represent averages obtained from three independent experiments. The list of transcripts was made based on data from the literature [4,11,31] and the KEGG database dedicated to the narrow-leaved blue lupin (*Lupinus angustifolius*) genome (https://www.genome.jp/kegg-bin/show_organism?menu_type=pathway_maps&org=lang; accessed on 16 June 2022). The transcriptomics data selected for the preparation of this figure are presented in Appendix A.

**Figure 6 ijms-24-11773-f006:**
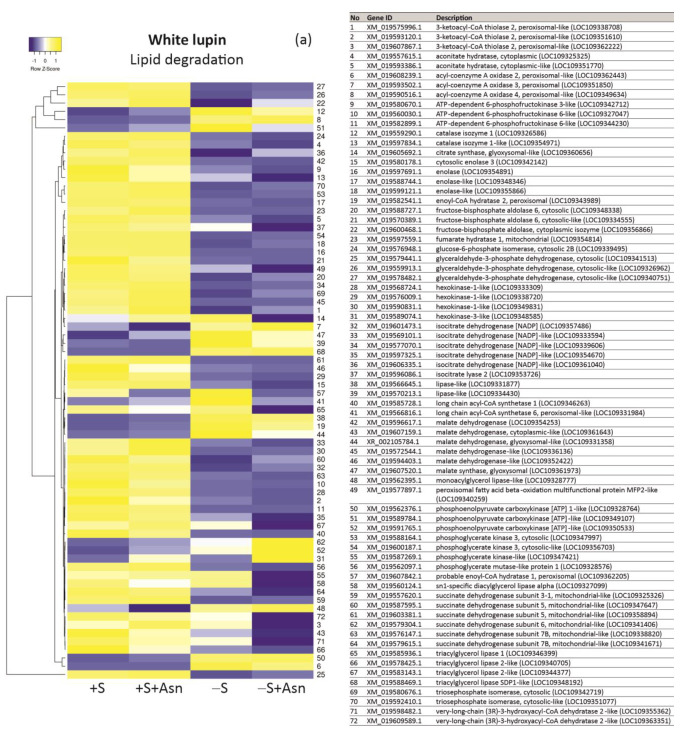
Heatmaps showing relations in transcript levels of genes encoding proteins involved in lipid degradation in white lupin (**a**) and Andean lupin (**b**) isolated embryonic axes cultured in vitro for 96 h on a medium with (+S) and without (-S) 60 mM sucrose. Culture media were also enriched with 35 mM asparagine (+Asn). The data represent averages obtained from three independent experiments. The list of transcripts was made based on data from the literature [2,9,10,11] and the KEGG database dedicated to the narrow-leaved blue lupin (*Lupinus angustifolius*) genome (https://www.genome.jp/kegg-bin/show_organism?menu_type=pathway_maps&org=lang; accessed on 14 June 2022). The transcriptomics data selected for the preparation of this figure are presented in Appendix A.

**Figure 7 ijms-24-11773-f007:**
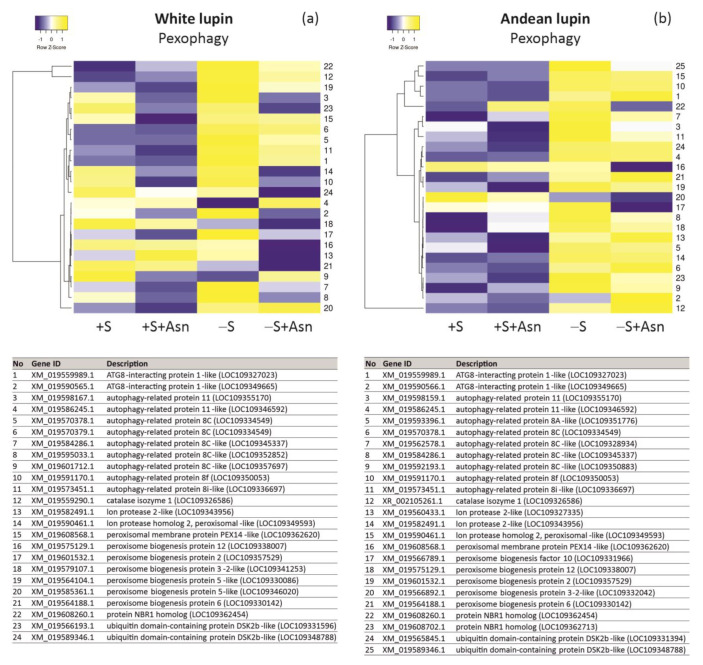
Heatmaps showing relations in levels of transcripts of genes encoding proteins involved in pexophagy (autophagic degradation of peroxisomes) in white lupin (**a**) and Andean lupin (**b**) isolated embryonic axes cultured in vitro for 96 h on a medium with (+S) and without (-S) 60 mM sucrose. Culture media were also enriched with 35 mM asparagine (+Asn). The data represent averages obtained from three independent experiments. The list of transcripts was made based on literature data [29,32,33,34]. Transcriptomics data selected for the preparation of this figure are presented in Appendix A.

**Figure 8 ijms-24-11773-f008:**
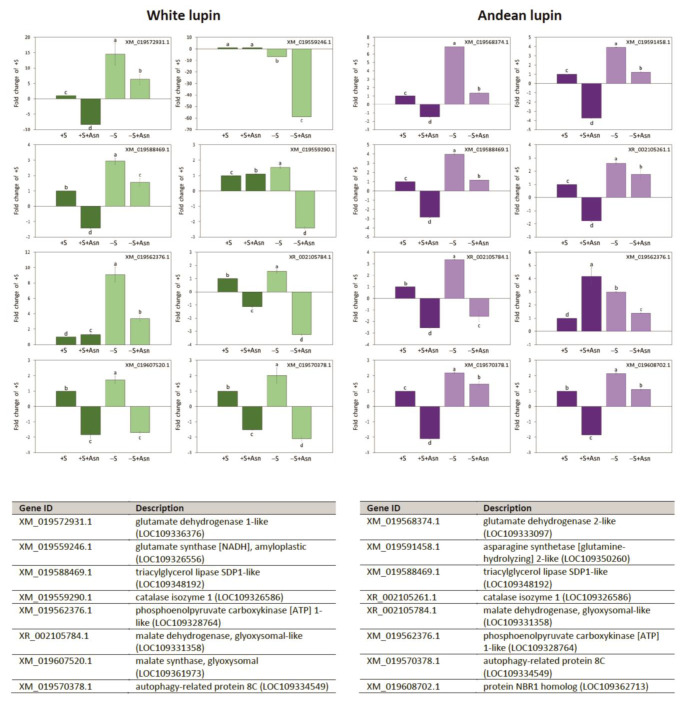
Relative changes in levels of selected gene transcripts determined by qRT-PCR technique in white and Andean lupin isolated embryonic axes cultured in vitro for 96 h on a medium with (+S) and without (-S) 60 mM sucrose. Culture media were also enriched with 35 mM asparagine (+Asn). Primer sequences used for qRT-PCR reactions are presented in Appendix A. The comparative C_T_ method for relative quantification was used with actin as an endogenous control. The amount of target, normalized to an endogenous reference and relative to the calibrator (+S; equal 1), is given by 2^−ΔΔCT^ [35]. The data represent averages obtained from three independent experiments. Different letters above the error bars (±SD) indicate statistically significant differences at *p* ≤ 0.05 (ANOVA, Tukey’s HSD multiple-range test).

**Figure 9 ijms-24-11773-f009:**
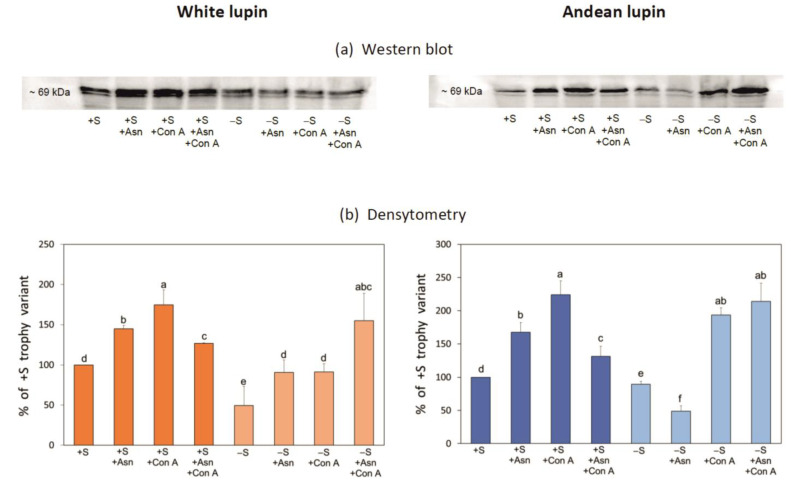
Western blot (**a**) and densitometric analysis (**b**) of the peroxisomal membrane protein Pex14 (Pex14p) content in white and Andean lupin isolated embryonic axes cultured in vitro for 96 h on a medium with (+S) and without (-S) 60 mM sucrose. Culture media were also enriched with 35 mM asparagine (+Asn). After 72 h of in vitro culture, for 24 h before the analysis, axes were transferred to media containing 10 μM concanamycin A (Con A), an inhibitor of autophagy (Con A inhibits the tonoplast PP_i_-dependent proton pump, causing neutralization of vacuolar pH, and thereby inhibits the activity of vacuolar hydrolytic enzymes, and in consequence, degradation of autophagic bodies inside the vacuole). The western blots presented in part (**a**) are representative of three independent experiments. Consistently, 40 μg of total protein was loaded per cell during electrophoresis. The data presented in part (**b**) are averages obtained from three independent experiments. The +S axes were set as 100% and were treated as a reference for axes from the rest of the trophic variants of the in vitro culture. Different letters above the error bars (±SD) indicate statistically significant differences at *p* ≤ 0.05 (ANOVA, Tukey’s HSD multiple-range test).

**Table 1 ijms-24-11773-t001:** Relations in protein levels determined by the iTRAQ method. The list includes proteins involved in lipid breakdown in embryonic axes isolated from white lupin germinating seeds and cultured in vitro for 96 h on a medium with (+S) and without (-S) 60 mM sucrose. Culture media were also enriched with 35 mM asparagine (+Asn). Protein quantification data were compared in the following pairs of axes: +S/+S+Asn, +S/-S, and -S/-S+Asn.

+S/+S+Asn				+S/-S				-S/-S+Asn				Protein ID/NCBI	Description White Lupin
*q*-Value	Ratio A/B	Fold Change	Peptide Number	*q*-Value	Ratio A/B	Fold Change	Peptide Number	*q*-Value	Ratio A/B	Fold Change	Peptide Number		
0.9588	1.04	1.04	6	0.0011	1.83	1.83	5	-	-	-	-	XP_019412859.1	long chain acyl-CoA synthetase 2-like
												XP_019420557.1	long chain acyl-CoA synthetase 2-like
0.9422	1.06	1.06	6	0.0000	0.55	1.82	11	0.6004	1.08	1.08	12	XP_019446061.1	acyl-coenzyme A oxidase 4, peroxisomal-like
1.0000	1.11	1.11	3	0.0017	0.47	2.11	5	1.0000	1.07	1.07	6	XP_019464789.1	acetate/butyrate-CoA ligase AAE7, peroxisomal-like
0.7892	0.99	1.01	6	0.0255	0.78	1.28	7	0.6844	1.05	1.05	4	XP_019453051.1	delta(3.5)-Delta(2.4)-dienoyl-CoA isomerase, peroxisomal isoform X1
1.0000	0.98	1.02	4	0.0029	0.67	1.5	8	1.0000	1.08	1.08	7	XP_019438086.1	enoyl-CoA hydratase, peroxisomal
1.0000	0.96	1.04	6	0.0405	0.79	1.26	7	1.0000	1.02	1.02	8	XP_019428389.1	3-ketoacyl-CoA thiolase 2, peroxisomal
1.0000	0.93	1.08	8	0.0013	0.74	1.34	12	1.0000	1.02	1.02	11	XP_019444968.1	malate dehydrogenase, glyoxysomal isoform X1
												XP_019444970.1	malate dehydrogenase, glyoxysomal isoform X2
1.0000	1.02	1.02	6	0.0354	0.77	1.3	10	1.0000	0.85	1.17	7	XP_019421327.1	malate dehydrogenase, glyoxysomal-like
1.0000	0.96	1.04	15	0.0004	0.8	1.25	19	0.0001	1.33	1.33	15	XP_019463065.1	malate synthase, glyoxysomal
1.0000	0.98	1.02	7	0.0951	0.79	1.27	8	0.0001	1.46	1.46	18	XP_019448948.1	isocitrate lyase 2-like
												XP_019451631.1	isocitrate lyase 2 isoform X1
												XP_019451632.1	isocitrate lyase 2 isoform X2
1.0000	1.02	1.02	11	0.0387	1.21	1.21	12	0.1246	1.15	1.15	9	XP_019435274.1	isocitrate dehydrogenase [NAD] catalytic subunit 5, mitochondrial
0.7401	1.08	1.08	15	0.0012	1.25	1.25	15	1.0000	0.98	1.02	12	XP_019436794.1	succinate-CoA ligase [ADP-forming] subunit alpha, mitochondrial
												XP_019457881.1	succinate-CoA ligase [ADP-forming] subunit alpha, mitochondrial-like
1.0000	1.03	1.03	28	0.0444	1.13	1.13	30	1.0000	0.93	1.08	29	XP_019459370.1	succinate-CoA ligase [ADP-forming] subunit beta, mitochondrial-like
												XP_019428639.1	succinate-CoA ligase [ADP-forming] subunit beta, mitochondrial-like isoform X1
												XP_019433054.1	succinate-CoA ligase [ADP-forming] subunit beta, mitochondrial
0.1451	0.90	1.11	3	0.0005	0.6	1.68	5	0.3194	0.85	1.18	9	XP_019417921.1	phosphoenolpyruvate carboxykinase [ATP] 1-like
0.1289	0.82	1.22	3	0.0001	0.51	1.96	7	1.0000	0.99	1.01	4	XP_019445329.1	phosphoenolpyruvate carboxykinase
-	-	-	-	0.0135	0.47	2.13	3	0.6389	1.27	1.27	2	XP_019420368.1	phosphoenolpyruvate carboxykinase [ATP]-like

Statistical analysis was performed with Diffprot software [30], implementing a non-parametric test for assessing significance, with correction for multiple testing. The differences in protein level reach statistical significance when the *q*-value is below 0.05. The green filling of the table cells indicates a statistically significant increase and the red filling of a cell indicates a statistically significant decrease (ratio A/B and fold change) in the protein level in axes cultured in vitro under different nutrient conditions. The bright grey filling of a cell indicates changes that did not reach statistical significance. The data represent results that were obtained from three independent experiments. The full iTRAQ data are presented in Appendix A.

**Table 2 ijms-24-11773-t002:** Relations in protein levels determined by the iTRAQ method. The list includes proteins involved in lipid breakdown in embryonic axes isolated from Andean lupin germinating seeds and cultured in vitro for 96 h on a medium with (+S) and without (-S) 60 mM sucrose. Culture media were also enriched with 35 mM asparagine (+Asn). Protein quantification data were compared in the following pairs of axes: +S/+S+Asn, +S/-S, and -S/-S+Asn.

+S/+S+Asn				+S/-S				-S/-S+Asn				Protein ID/NCBI	Description Andean Lupin
*q*-Value	Ratio A/B	Fold Change	Peptide Number	*q*-Value	Ratio A/B	Fold Change	Peptide Number	*q*-Value	Ratio A/B	Fold Change	Peptide Number		
0.3273	1.07	1.07	12	0.0199	0.82	1.22	14	0.1969	1.21	1.21	14	XP_019449047.1	acyl-coenzyme A oxidase 3, peroxisomal
-	-	-	-	0.9065	0.98	1.02	3	0.0123	0.59	1.69	6	XP_019463387.1	probable enoyl-CoA hydratase, peroxisomal
1.0000	1.09	1.09	7	0.4980	0.90	1.11	4	0.0014	0.47	2.12	7	XP_019453379.1	catalase isozyme 1-like
1.0000	0.96	1.05	13	0.1186	1.22	1.22	14	0.0004	0.65	1.53	14	XP_019449188.1	malate dehydrogenase, glyoxysomal
1.0000	1.16	1.16	16	0.0001	0.62	1.62	17	0.0001	1.65	1.65	19	XP_019435367.1	citrate synthase, glyoxysomal-like
												XP_019461237.1	citrate synthase, glyoxysomal-like
0.7026	0.92	1.08	14	0.0008	0.68	1.46	9	0.4166	1.17	1.17	7	XP_019451631.1	isocitrate lyase 2 isoform X1
												XP_019451632.1	isocitrate lyase 2 isoform X2
1.0000	1.08	1.08	11	1.0000	1.04	1.04	2	0.0166	1.35	1.35	11	XP_019423390.1	aconitate hydratase 1
												XP_019444853.1	aconitate hydratase 1-like isoform X2
1.0000	0.97	1.03	6	0.0477	0.66	1.53	4	0.0807	0.78	1.28	9	XP_019417921.1	phosphoenolpyruvate carboxykinase [ATP] 1-like

Statistical analysis was performed with Diffprot software [30], implementing a non-parametric test for assessing significance, with correction for multiple testing. The differences in protein level reach statistical significance when the *q*-value is below 0.05. The green filling of the table cells indicates a statistically significant increase and the red filling of a cell indicates a statistically significant decrease (ratio A/B and fold change) in the protein level in axes cultured in vitro under different nutrient conditions. The bright grey filling of a cell indicates changes that did not reach statistical significance. The data represent results obtained from three independent experiments. The full iTRAQ data are presented in Appendix A.

## Data Availability

The mass spectrometry proteomics data for white and Andean lupin embryonic axes have been deposited to the ProteomeXchange Consortium via the PRIDE [64] partner repository with the dataset identifier PXD041380, Project doi:10.6019/PXD041380, https://www.ebi.ac.uk/pride/, deposited on 6 April 2023. Transcriptomics data (NGS) have been deposited to the SRA database. The BioProject accession number for white lupin: PRJNA953600, https://www.ncbi.nlm.nih.gov/sra/PRJNA953600, deposited on 11 April 2023, and Andean lupin: PRJNA953433, https://www.ncbi.nlm.nih.gov/sra/PRJNA953433, deposited on 9 April 2023.

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
