# Peer review of "Sugar Starvation Disrupts Lipid Breakdown by Inducing Autophagy in Embryonic Axes of Lupin (*Lupinus* spp.) Germinating Seeds"

_ijms, 2023, doi:10.3390/ijms241411773_

Round 1
Reviewer 1 Report
The author conducted a prospective exploration on the relationship between early lipid metabolism and autophagy in lupin beans, which is interesting.
But the author still needs to clarify:
1. How to distinguish Autophagosome from autolyst under electron microscope. Autophagosome usually contains a double membrane structure. It is not shown in Figure 2, please zoom in further.
2. The necessary control protein is missing in Figure 9a.
Author Response
The author conducted a prospective exploration on the relationship between early lipid metabolism and autophagy in lupin beans, which is interesting.
We are thankful to Reviewer for a positive general opinion.
But the author still needs to clarify:
- How to distinguish Autophagosome from autolyst under electron microscope. Autophagosome usually contains a double membrane structure. It is not shown in Figure 2, please zoom in further.
Yes, autophagosomes are surrounded by a double-double-layer phospholipid membrane. They occur only in the cytoplasm. But autophagic bodies are surrounded only by a single double-layer phospholipid membrane and they occur only inside the vacuole. Nevertheless, in our research, we did not focus on autophagosomes. Only autophagic bodies, which result from autophagosome fusion with the vacuole and exist only inside vacuoles were of our interest during this research. Autophagosomes will be one of the objects of our future work, where we are going to present the results of autophagosome staining with monodansylcadaverine under a confocal microscope.
Frankly speaking, we do not understand the term “autolyst” mentioned by the reviewer. We can only suppose that it should be an “autolysosome”. If yes, such structures are surrounded by only a single double-layer phospholipid membrane but they occur only in animal cells, so are not related to our research at all.
- The necessary control protein is missing in Figure 9a.
We are grateful to the Reviewer for this remark because we realized that we have omitted very important information in Materials and Methods. Always 40 micrograms of total protein was loaded per cell during electrophoresis. In the densitometric analysis, the +S axes were always set as 100% and were treated as a reference for axes from the rest of the trophic variants of the in vitro culture. We added missing information to the legend of Figure 9 (lines 440-443 of the revised manuscript) and the Materials and Methods (line 789).

Reviewer 2 Report
Very inetersting paper.
I would suggest to look detaily on the chromatin organization. and atveast discuss these points.
Line 46: “Mature and dry lupin seeds are non-starch seeds”?
Figure 1: not clear. Please, show origin of phosphoenolpyruvate as source of glucose/sucrose. In the current version it somehow confusing Please, have a look:
https://doi.org/10.1016/j.pbi.2007.04.007
Line 142: “to provide evidence on pexophagy in cells”? Do you mean presence or functionality or activity?
Lines 143-147: please, formulate your hypothesis and task instead of describing your observation.
Figure 2: is very questionable. Meristem zone means by definition actively proliferating cells. Usually, carbon starvation lead to significant changes in cell struhttps://doi.org/10.1101/353987ture and cell division activity. Authors have to provide prove that they investigate meristem cell. Moreover, cell structure is also cell-type dependent. Are you sure that you compare the same cell types? As I have seen, the most interesting effect of starvation is on nuclei size. This should be described in next paper and mention in discussion in the current one.
https://doi.org/10.1101/353987
Figure 3: Panel A: quality is very low, probably through screen copy. It is not so easy to recognize colors.
Panel B: low quality of the graphs, maybe after pdf conversion.
Line 196: “axes” – redundant.
with 60 mM sucrose (+S) or without sugar (-S)” = “with and without sucrose”. You do need to repeat always 60 mM sucrose, since you have used only one concentration.
Line 314: what about other enzymes “clear decrease in the level of the majority of transcripts” – what about other less specific transcripts? What about root development and cell cycle transcripts?
Line 345 “with 60 mM sucrose (+S) or without sugar (-S)” – redundant!
Line 633: “distilled and autoclaved water” = sterile water.
Line 634: “Embryonic axes” some illustration and explanation what do you consider as embryonic axis will be useful.
Line 635: “above” = on; “with 60 mM sucrose (+S) and without sucrose (-S)” = with (+S) and without (-S) 60 mM sucrose.
Line 636: “Asparagine solutions:::” – please. Provide details: concentration, pH. Addition of asparagine may changes the concentration of other components as well as pH. How this have been considered?
Line 637: “by passing through” = with.
Line 643: in which buffer?
“4% paraformaldehyde” is non-sense. Para mean a powder! Once you dissolve paraformaldehyde it became a 4% formaldehyde!
Lines 768: how do you normalize WB? Per total protein? Or per cell? Please, provide details.
corrections required!
Author Response
Very inetersting paper.
We are thankful to Reviewer for such a positive attitude to our manuscript.
I would suggest to look detaily on the chromatin organization. and atveast discuss these points.
Sugar starvation causes many ultrastructural changes not only in cells of the root meristematic zone. We described these changes in several of our previous papers, like Borek et al. 2002 [20], 2006 [22], 2011 [11], 2012 [19], 2013 [3], and 2017 [8], or Morkunas et al. 2012 [23] (all these papers are cited in the manuscript and appropriate references’ numbers of the revised manuscript are provided above). During the research described in these papers, we also looked at nuclei structure. In general, nuclei in cells of the root meristematic zone of sucrose-fed lupin axes (+S) are visible in the majority of cells, often are very large, and fill the majority of the area of a cell cross-section. Also, nucleoli are often clearly visible, and no or rarely euchromatin is visible. The nuclear envelope is smooth without any damage. But under sugar starvation conditions (-S axes) the vacuole becomes very large and this causes nuclei to be not as frequently visible as in +S axes. Nuclei are often shifted to the peripheral part of the cross-section of a cell, close to the cell wall. Fewer nucleoli are visible, more euchromatin can be detected, and the nuclei envelope is waved. However, all these changes are already described in our earlier papers, and the cell nucleus is not a key element of this paper, so we would like not to rise this point in this manuscript, afraid that the main object of the manuscript, i.e., the lipid breakdown during lupin seed germination, may become unclear.
Line 46: “Mature and dry lupin seeds are non-starch seeds”?
Yes, mature and dry lupin seeds contain only trace amounts of starch so are considered non-starch seeds. However, starch is present in lupin seeds during development and maturation. Its content decrease during lupin seeds maturation and desiccation, and is almost completely despaired in mature and dry lupin seeds. Only traces of starch content was detected in lupin dry seeds (Borek et al. 2013) [3]. But starch is newly synthesized during the first hours of lupin seed imbibition and germination, and it is a kind of transitory starch, which again disappeared during further stages of lupin seed germination and seedling growth. We have described several times the changes in the starch presence (ultrastructure) or the content (analytical measurement) in developing and germinating lupin seeds in some of our previous papers, like Borek et al. 2009 [53], 2011 [11], 2013 [3], or 2015 [2]. Our paper by Borek et al. 2013 [3] fully is about starch metabolism in lupin seeds and we pointed to starch as an important component of a sink-source (donor-acceptor) system creation and maintenance in lupin seeds. Due to the Reviewer's question, we slightly modified the Introduction (lines 46-51) and added more details about the starch content in lupin seeds. We add there a new reference. Namely, we cite there our paper by Borek et al. 2013 and added it to the References [3]. This new information will be also necessary to explain another question of the Reviewer, namely the question about enzyme activity.
Figure 1: not clear. Please, show origin of phosphoenolpyruvate as source of glucose/sucrose. In the current version it somehow confusing Please, have a look:
https://doi.org/10.1016/j.pbi.2007.04.007
The paper suggested by the Reviewer concerns only the b-oxidation of fatty acid, but the source of phosphoenolpyruvate as a beginning molecule for gluconeogenesis is not the b-oxidation directly. Phosphoenolpyruvate is generated from oxaloacetate in the cytoplasm by the action o phosphoenolpyruvate carboxykinase (PEPCK), and there are two sources of oxaloacetate. One is the glyoxylate cycle. Oxaloacetate then is synthesized in the cytoplasm by the action of cytosolic malate dehydrogenase (cMDH). The second source of oxaloacetate is the TCA cycle. Please refer to Borek et al. 2015 [2] and the literature cited therein. Both of these two sources of oxaloacetate for phosphoenolpyruvate synthesis by PEPCK are marked in Figure 1. However, we suppose, that the arrows leading from these two oxaloacetate sources to phosphoenolpyruvate maybe are not as clear as possible, so we corrected slightly the bottom left area in Figure 1.
Line 142: “to provide evidence on pexophagy in cells”? Do you mean presence or functionality or activity?
Indeed, this wording was not clear, so we deleted this part of the sentence (lines 146-147).
Lines 143-147: please, formulate your hypothesis and task instead of describing your observation.
Please excuse me, but we did not provide any observations or results in this place of the manuscript. Already in the original version of the manuscript, in this place, we had written the tasks we were going to do. There are none of our observations there. Thus, we do not introduce any changes in this place, because this part of our manuscript already was in agreement with the Reviewer's recommendation (lines 147-151 of the revised manuscript).
Figure 2: is very questionable. Meristem zone means by definition actively proliferating cells. Usually, carbon starvation lead to significant changes in cell struhttps://doi.org/10.1101/353987ture and cell division activity. Authors have to provide prove that they investigate meristem cell. Moreover, cell structure is also cell-type dependent. Are you sure that you compare the same cell types? As I have seen, the most interesting effect of starvation is on nuclei size. This should be described in next paper and mention in discussion in the current one.
https://doi.org/10.1101/353987
Yes, we are sure that we present cells of the root meristematic zone of lupin embryonic axes in Figure 2. We have many-year experience in ultrastructure observations. For the first time, I have prepared roots for transmission electron microscope (TEM) already in 90-ty of the past century. To be sure we observe cells of the root meristematic zone cells we first start to cut the axes embedded in the epoxy resin from the tip of a root cap. Step by step, we make a series of semi-thin (2.5 micrometers) cross-sectioned specimens and analyzed them first under the light microscope. Additionally, always we count these microscopy sections to successively build our consciousness and knowledge about the structure of the root apex of our embryonic axes. For the ultrastructural observations of the root meristematic zone under a TEM, we choose only the level where inside of the cross-section of the root apex small and closely packed cells are visible and this core is surrounded by cells of the root cap. These root cap cells are clearly distinguishable from the central meristematic zone cell. They are bigger, lossy packed, contain large starch granules, are flacking off, and often a mucus is visible around a section. Additionally, if we are too shallow in the series of semithin specimens, we see under the light microscope cells of a root cap, particularly columella cells, with clear and large granules of starch statoliths, but when we are too deep, the core of the root cross-section is composed with much bigger cells, they are not the as closely packed ad in the meristematic zone and they are not surrounded by flaking off root cap cells. Even though we first orientated the embedded in epoxy resin embryonic axes under the light microscope before the TEM observations, we also do a recognizance under the TEM before the proper observations. Thus we are sure that we observe cells of the root meristematic zone.
To avoid such doubts as the Reviewer riced, we add a short information to the Materials and Methods about the orientation of the roots for ultrastructure observations (lines 658-664).
Figure 3: Panel A: quality is very low, probably through screen copy. It is not so easy to recognize colors.
Yes, we agree that the resolution of all figures in the PDF file is weak, but it is not a screen copy. We can assure the reviewer that in the original figure files, attached along with the submission, the resolution is considerably higher and all details are visible and distinguishable. Nevertheless, the Reviewer’s remark has caused us to enlarge the keys of all points and make the lines thicker in Figure 3a to make them more readable and user-friendly.
Panel B: low quality of the graphs, maybe after pdf conversion.
Yes, the low resolution is a result of Word into PDF conversion. However, we assure the original files of all figures are of remarkably higher quality than what is visible in the PDF file.
Line 196: “axes” – redundant.
Corrected. We removed the redundant word.
Lines 206-231: Enzyme activity and protein itself require carbon as structural component. In starch-free seeds carbon source very limited and maybe this is the main reason of the events described? Please, discuss and mention this.
Only dry lupin seeds are non-starch seeds. During the first hours of seed imbibition, storage compounds, such as storage proteins (the main carbon source in lupin germinating seeds; up to 50% of lupin seed dry mater is a storage protein; Borek et al. 2015 [2]), storage lipid (up to 20%; Borek et al. 2015 [2]) or oligosaccharides (up to about 36%; Borek et al. 2015 [2]) are mobilized and a supply of respiratory and anabolic substrates take place. The supply of these metabolites probably is even too high for the demand in the source-sink system, and the above-mentioned transitory starch quickly appears during lupin seed imbibition and germination. We described and discussed this topic in one of our earlier papers by Borek et al. 2013 [3]. Thus, regarding the Reviewer's question, we can answer, that the starch-dependent carbon supply or carbon deficit is not a reason for a picture in enzyme activity changes. First of all, sucrose has a meaning, because this soluble sugar is a regulatory agent in plant metabolism. Sucrose, as well as glucose and trehalose, can affect the expression of hundreds or even thousands of genes in plants. We described in detail this topic in our book chapter by Morkunas et al. 2012 [23].
Answering one of the earlier questions from the Reviewer, we already slightly modified the Introduction and added a new reference regarding starch content in lupin seed (Borek et al. 2013 [3], lines 46-51) and we would like not more discuss this topic in the manuscript. We are afraid that we dilute the main idea of the manuscript.
Line 235: “with 60 mM sucrose (+S) or without sugar (-S)” = “with and without sucrose”. You do need to repeat always 60 mM sucrose, since you have used only one concentration.
We are thankful for such advice. We corrected it in all places in the manuscript as possible. Mainly in figure legends, table headings, Materials and methods, as well in supplementary files.
Line 314: what about other enzymes “clear decrease in the level of the majority of transcripts” – what about other less specific transcripts? What about root development and cell cycle transcripts?
Our NGS gave us 57.264 records for each lupin species. To maintain a clear main idea of the manuscript we had to focus on selected points in lupin seed metabolism. We decided to examine genes of enzymes involved in amino acid metabolism (because lupin seeds are very rich in storage protein), but the main topic of our manuscript is lipid catabolism, so we selected genes related to katabolism of this storage compound, as well as on genes related to pexophagy. Originally we also planned to present heatmaps related to peroxisome, but we decided finally to not present such data because it would be a negative effect on the main message of our manuscript. You mentioned root development and cell cycle transcripts, but we are afraid that more NGS data added to our manuscript may cause the main subject to blur. Nevertheless, we are thankful for such a suggestion and we will remember for sure your advice during the preparation of further manuscripts.
Line 345 “with 60 mM sucrose (+S) or without sugar (-S)” – redundant!
As we mentioned above, we corrected it in several places in the manuscript.
Line 633: “distilled and autoclaved water” = sterile water.
Corrected accordingly (line 643).
Line 634: “Embryonic axes” some illustration and explanation what do you consider as embryonic axis will be useful.
Description and photos of lupin embryonic axes we used in our research, were already published earlier, for example in Borek et al. 2012 or 2017. We added already cited Borek et al. 2017 [8] to the Material and Methods and slightly modified the description of plant material (lines 650-651).
Line 635: “above” = on; “with 60 mM sucrose (+S) and without sucrose (-S)” = with (+S) and without (-S) 60 mM sucrose.
Corrected as described above.
Line 636: “Asparagine solutions:::” – please. Provide details: concentration, pH. Addition of asparagine may changes the concentration of other components as well as pH. How this have been considered?
We are thankful for this remark because we realized that made a mistake in Materials and Methods. We did not make separate asparagine solutions but added the appropriate asparagine amounts directly to +S or -S media. Thus, the enrichment of media with asparagine could not dilute other ingredients of the Heller medium and we also do not record any pH changes after asparagine adding. The enriched with asparagine media were next sterilized by Millipore syringe filters. We corrected the description in the Materials and Methods (lines 647-649).
Line 637: “by passing through” = with.
Corrected accordingly (line 648).
Line 643: in which buffer?
Cacodylate buffer, 0.05 M, pH 6.8. We added this information to the description in the Materials and Methods (line 656).
“4% paraformaldehyde” is non-sense. Para mean a powder! Once you dissolve paraformaldehyde it became a 4% formaldehyde!
Yes, of course. We used a ready-made solution of formaldehyde provided by Polysciences. We corrected it in the Materials and Methods (line 656).
Lines 768: how do you normalize WB? Per total protein? Or per cell? Please, provide details.
Per total protein in a sample. It always was 40 micrograms per cell during electrophoresis. We already provided a more detailed explanation in response to Reviewer 1 remark and introduced appropriate corrections to the manuscript (lines 440-443 and line 789).

Round 2
Reviewer 2 Report
Thank you very much for the response.
It will be great if you show full view (overview) of the root apex under different conditions and put some of yur image in the contents of the root apex. and insert at least sIn teh case of root apex cell with different fate/position have completely different structure and randonmly selected cell can not be compared. Please, add some of the statement you add to the comments to main text. It seems that you use only self-citations. It will be nice to add similar one from the other authors.
Kircher, S., & Schopfer, P. (2012). Photosynthetic sucrose acts as cotyledon-derived long-distance signal to control root growth during early seedling development in Arabidopsis. Proceedings of the National Academy of Sciences, 109(28), 11217-11221.
The figure 1 as well as other graphs are clear now. Thanks!
OK
Author Response
Reviewer 2
Thank you very much for the response.
It will be great if you show full view (overview) of the root apex under different conditions and put some of yur image in the contents of the root apex.
We added to the manuscript Supplementary Figure S1 where we present the morphology of the isolated embryonic axes used in our research (Figure S1a) and precisely showed which area of the root apex was observed under a transmission electron microscope (Figure S1b). We corrected section 4.1. Plant material by introducing there the information about Figure S1a (line 651). Information about the detailed description and localization of the ultrastructure observations area (Figure S1ab) we also added to section 4.2. Ultrastructure (lines 665-667).
and insert at least sIn teh case of root apex cell with different fate/position have completely different structure and randonmly selected cell can not be compared.
We left this comment without any action because the beginning is not understandable to us and we do not know what we should correct. We can only ensure that no randomly selected cells were compared by us.
Please, add some of the statement you add to the comments to main text.
During the first round of the review, we provided a 7-page cover letter with many of our responses to reviewers' comments. Unfortunately, the Reviewer does not indicate now, which statements we should move from our comments to the main text. Under such circumstances, we decided to move only our comment about the ultrastructure to the legend of Figure S1, and there we describe in detail the localization of the area of ultrastructure observations.
It seems that you use only self-citations. It will be nice to add similar one from the other authors.
Indeed, we cited several of our papers, but it must be clearly emphasized, that there are no other papers about carbohydrate and lipid metabolism in lupin germinating seeds than ours. This is the reason for self-citation.
Kircher, S., & Schopfer, P. (2012). Photosynthetic sucrose acts as cotyledon-derived long-distance signal to control root growth during early seedling development in Arabidopsis. Proceedings of the National Academy of Sciences, 109(28), 11217-11221.
Thank you for this hint. We added this paper to the manuscript. Number [40], line 450.
doi: https://doi.org/10.1101/353987
We are sorry but we did not add this paper to our manuscript. Firstly, it does not match the topic of our manuscript, and the citation of such a paper would be not correct. Secondly, it is only a pre-print from the 2018 year, and we can not find the final published paper.
However, the reviewer’s remark has encouraged us to add two other references. They are Kawamata et al. 2022 [27] line 136, and Tyutereva et al. 2022 [44] line 573.
The figure 1 as well as other graphs are clear now. Thanks!
Thank you very much.
Comments on the Quality of English Language
OK
Thank you very much.
Round 3
Reviewer 2 Report
Thank you! The paper is fine now. Dzięki !
ok